



# Interannual and seasonal variability of the air–sea CO$_2$ exchange at Utö in the coastal region of the Baltic Sea

Martti Honkanen[1], Mika Aurela[2], Juha Hatakka[2], Lumi Haraguchi[3], Sami Kielosto[3], Timo Mäkelä[2], Jukka Seppälä[3], Simo-Matti Siiriä[1], Ken Stenbäck[1], Juha-Pekka Tuovinen[2], Pasi Ylöstalo[3], and Lauri Laakso[1,4]

[1]Meteorological and Marine Research Programme, Finnish Meteorological Institute, Helsinki, Finland
[2]Climate Research Programme, Finnish Meteorological Institute, Helsinki, Finland
[3]Research Infrastructure Unit, Finnish Environment Institute, Helsinki, Finland
[4]School of Physical and Chemical Sciences, North-West University, Potchefstroom Campus, South Africa

**Correspondence:** Martti Honkanen (martti.honkanen@fmi.fi)

**Abstract.** Oceans alleviate the accumulation of atmospheric CO$_2$ by absorbing approximately a quarter of all anthropogenic emissions. In the deep oceans, carbon uptake is dominated by aquatic phase chemistry, whereas in biologically active coastal seas the marine ecosystem and biogeochemistry play an important role in the carbon uptake. Coastal seas are hotspots of organic and inorganic matter transport between the land and the oceans, and thus important for the marine carbon cycling. In this study,

we investigate the net air–sea CO$_2$ exchange at the Utö Atmospheric and Marine Research Station, located at the southern edge of the Archipelago Sea within the Baltic Sea, using the data collected during 2017–2021. The air–sea fluxes of CO$_2$ were measured using the eddy covariance technique, supported by the flux parametrization based on the $p$CO$_2$ and wind speed measurements. During the spring–summer months (April–August), the sea was gaining carbon dioxide from the atmosphere, with the highest monthly sink fluxes typically occurring in May, being -0.26 μmol m$^{-2}$ s$^{-1}$ on average. The sea was releasing

the CO$_2$ to the atmosphere in September–March, and the highest source fluxes were typically observed in September, being 0.42 μmol m$^{-2}$ s$^{-1}$ on average. On the annual basis, the study region was found to be a net source of atmospheric CO$_2$, and on average, the annual net exchange was 27.1 gC m$^{-1}$ y$^{-1}$, which is comparable to the exchange observed in the Gulf of Bothnia, the Baltic Sea. The annual net air–sea CO$_2$ exchanges varied between 18.2 gC m$^{-1}$ y$^{-1}$ (2018) and 39.1 gC m$^{-1}$ y$^{-1}$ (2017). During the coldest year, 2017, the spring–summer sink fluxes remained low compared to the other years, as a result of relatively

high seawater $p$CO$_2$ in summer, which never fell below 220 μatm during that year. The spring–summer phytoplankton blooms of 2017 were weak, possibly due to the cloudy summer and deeply mixed surface layer, which restrained the photosynthetic fixation of dissolved inorganic carbon in the surface waters. The algal blooms in spring–summer 2018 and the consequent $p$CO$_2$ drawdown were strong, fueled by high pre-spring nutrient concentrations. The systematic positive annual CO$_2$ balances suggest that our coastal study site is affected by carbon flows originating from elsewhere, possibly as organic carbon which

is remineralized and released to the atmosphere as CO$_2$. This coastal source of CO$_2$ fueled by the organic matter originating probably from land ecosystems stresses the importance of understanding the carbon cycling in the land-sea continuum.



# 1 Introduction

Carbon dioxide ($CO_2$) is the key greenhouse gas driving the global climate change and ocean acidification (Feely et al., 2009).
The $CO_2$ exchange between the atmosphere and land ecosystems has been extensively studied during the recent decades
(Baldocchi et al., 2001; Pastorello et al., 2020). However, to fully understand the global carbon cycling, it is crucial to recognise
the role of the oceans, which absorb 26% of the anthropogenic $CO_2$ emissions (Friedlingstein et al., 2022). Direct measurements
of the atmospheric $CO_2$ concentrations and the terrestrial ecosystem fluxes have long traditions across the Earth (Keeling et
al., 1976; Baldocchi et al., 2001; Heiskanen et al., 2022), while the logistical and technical challenges have hindered the
measurements in the oceans until the last few decades (Bakker et al., 2016; Sloyan et al., 2019).

Approximately half of the net primary production in the Earth takes place in the oceans (Field et al., 1998), which cover 71
% of the Earth's surface. Thus, they are globally the largest ecosystem contributing to the global carbon balance. Solubility and
biological pumps effectively transport carbon to the deeper ocean (Volk and Hoffert, 1985). The solubility pump is governed by
the large scale oceanic circulation which transports water masses to the polar regions, where these water masses cool, absorb
atmospheric $CO_2$ due to increased solubility and sink into the deeper ocean. The biological pump is related to the synthesis
of organic carbon in the surface waters and its mobilization to the deeper ocean. On average, the oceans are sinks of $CO_2$, but
the air–sea $CO_2$ exchange is highly variable (Takahashi et al., 2002). In coastal areas, large amount of matter is transported
between the land, the oceans and the atmospheres. Approximately 40% of the carbon sequestration through the biological
carbon pump occurs in the coastal seas (Muller-Karger et al., 2005).

The $CO_2$ exchange between the atmosphere and the sea ($FCO_2$) is driven by the $CO_2$ partial pressure ($pCO_2$) difference
between the sea and the atmosphere (sea - atmosphere), and can be expressed using the temperature-salinity driven solubility
factor of the $CO_2$ ($K_0$) and the gas transfer velocity ($k$):

$$FCO_2 = kK_0\Delta pCO_2 \tag{1}$$

Thus, the convention used here is the positive fluxes denoting exchange from the sea to the atmosphere. Several factors affect
the gas transfer velocity and typically it has been parameterized using wind speed (Wanninkhof, 1992; Wanninkhof et al., 2009;
Wanninkhof, 2014).

The Baltic Sea is one of the biggest brackish water basins in the world. This shallow sea, with a mean depth of 54 m, has
strong seasonality in temperature (approximately 0–20 °C) and in primary production, being affected by large terrestrial input
of organic and inorganic carbon (Räike et al., 2012). Eutrophication is one of the biggest environmental concerns for the Baltic
Sea (Andersen et al., 2017), resulting in harmful algae blooms (Kahru and Elmgren, 2014; Kraft et al., 2021) and oxygen
depletion in deep waters (Carstensen et al., 2014). In the Baltic Sea, the $pCO_2$ in the seawater is regulated by the interplay
between biological, chemical and physical processes (Omstedt et al., 2009). The net $CO_2$ balance for the whole Baltic Sea
has been estimated to be close to neutral (Kuliński et al., 2014; Ylöstalo et al., 2016), but the estimates for its different basins





show large variation. Typically, the southern parts of the Baltic Sea act as a sink of $CO_2$ due to their strong primary production
(Kuliński and Pempkowiak, 2011; Kuss et al., 2006). Whereas the northern part has lower primary production and receives a
large amount of organic matter which remineralizes into inorganic carbon, making this area a source of $CO_2$ (Algesten et al.,
2006).

In addition to the spatial variability, the net air–sea exchange of $CO_2$ at a given location in the Baltic Sea experiences also
large seasonal variability (Rutgersson et al., 2008), and likely also interannual variability. Our understanding of these variations
is still incomplete, and understanding the physical, biological and chemical constraints on the Baltic Sea carbon cycle requires
sustained high-frequency measurements. By using daily FerryBox $pCO_2$ data, Schneider et al. (2014) found the central Baltic
Sea to act as a weak sink of the atmospheric $CO_2$, the annual exchange varying within -7 – -11 gC m$^{-2}$ y$^{-1}$. In contrast,
Wesslander et al. (2010) used long data series of monthly monitoring data of total alkalinity and pH to calculate the seawater
$pCO_2$ and the air–sea $CO_2$ fluxes and found the central Baltic Sea to mainly act as a source of atmospheric $CO_2$, the annual
balance varying from close to zero to up to almost 50 gC m$^{-2}$ y$^{-1}$.

The calculation of the air–sea $CO_2$ exchange is typically based on the $pCO_2$ data and wind speed dependent parametrization
of the gas transfer velocity (Eq. 1). For the coastal seas, direct high frequency measurements are needed for the reliable
estimation of the net $CO_2$ exchange at a given location. For example, monthly cruises can be too infrequent to detect the rapid
variations of $pCO_2$ in the Baltic Sea (Kuss et al., 2006; Honkanen et al., 2021). Also, the calculation of the seawater $pCO_2$
from other carbonate system variables introduces some uncertainty in the $pCO_2$ estimate (Steinhoff, 2020) The eddy covariance
technique (EC) provides means for directly measuring the instantaneous fluxes of matter and energy between the surface and
the atmosphere. However, long-term measurements are technically challenging in a marine environment, and the EC method
has been applied only at few locations in the Baltic Sea for measuring the air–sea $CO_2$ fluxes. Currently, continuous long-term
air–sea $CO_2$ EC flux measurements take only place at Östergarnsholm (Rutgersson et al., 2008) and Utö (Honkanen et al.,
75  2018).

The aim of this study was to determine the annual $CO_2$ net exchange between the sea and the atmosphere and its seasonal
and interannual variability in the Archipelago Sea, a part of the Baltic Sea. Time series of the EC flux data, supported with
parameterized gas transfer velocity and measured $pCO_2$ gradient, gathered at Utö within 2017–2021 were used for the deter-
mination of the net exchange of $CO_2$. Physical and biological measurements provide background to explain the reasons for
seasonal and annual variations in the flux data.

## 2   Materials and methods

### 2.1   Study site

The Utö atmospheric and marine research station (Fig. 1) is located at the outer edge of the Archipelago Sea of the Baltic
Sea (59°46′ 55" N, 21°21′ 27" E). The Archipelago Sea is bordered by the southwestern coast of Finland, the Åland Sea, the
Bothnian Sea and the Baltic Proper. This area comprises thousands of small islands and provides with an important gateway
for the water exchange between the Baltic Proper and the Bothnian Sea (Miettunen et al., 2023). Utö is the southernmost



permanently inhabited island of Finland, with some dozens permanent inhabitants throughout the year. The vegetation on this small rocky island (0.8 km$^{-2}$) with few trees is mainly bushes and shrubs (Kilkki et al., 2015), especially on the western side of the island. Southwest of the island, the sea quickly deepens to 80 m (Honkanen et al., 2018), starting the open seas of the the Northern Baltic Proper. Generally, temperature in the surface waters at Utö varies within 0–18 °C (Laakso et al., 2018). Sea ice is observed at Utö every few years (Jevrejeva et al., 2004; Laakso et al., 2018).

**Figure 1.** Location, map and scientific measurements of Utö, which include (1) Marine research station and the micrometeorological flux tower next to it (2) Optical temperature fibre tower, (3) The Integrated Carbon Observing System (ICOS) Atmosphere station, (4) X-band radar, (5) Atmosphere research station, (6) Acoustic Doppler Current Profiler, (7) conductivity-temperature-depth vertical profiling, (8) Seawater thermistor chain, (9) Profiling buoy and (10) Submerged pump system.

The island of Utö has a long history of measurements, as the earliest meteorological and hydrographic observations date back to 1881 and 1900, respectively (Laakso et al., 2018). The Utö atmospheric and marine research station is a comprehensive atmospheric and marine observing system combining continuous long-term observations of the atmosphere, sea physical state, biogeochemistry and marine ecosystem dynamics. Recently, the site has been developed also towards a marine radar and



radio signal research supersite (Rautiainen et al., 2023). The Utö station, runs in co-operation between the Finnish Meteorological Institute and the Finnish Environment Institute, comprising a variety of research facilities within and around the Utö island. It is part of the Joint European Research Infrastructure for Coastal Observatories (JERICO), national marine research Infrastructure FINMARI, and recently the Aerosol, Clouds and Trace Gases Research Infrastructure (ACTRIS). Also, there
is an Integrated Carbon Observing System (ICOS) Atmosphere station, measuring atmospheric carbon constituents, on the island. The observing system used in the study is described in the next subsections. In this study, we used the 30 min averaged observations relevant for the air–sea $CO_2$ exchange during 2017–2021.

## 2.2 Flow-through system

Continuous observations of seawater properties are based on a flow-through system, similar to those on many FerryBox systems
(Rantajärvi et al., 2003; Petersen et al., 2011), with a large number of parameters measured (Fig. 2). A submerged borehole pump (Grundfors SP3A-9N) is located approximately 250 m west of the island at the bottom of the sea, at the depth of 23 m $\pm$ 0.5 m (Honkanen et al., 2021; Kraft et al., 2021), next to an Acoustic Doppler Current Profiler and a thermistor chain. A sample water inlet system, equipped with a thermistor-depth sensor, is floating above the pump at the depth of 4.5 m $\pm$ 0.5 m. The flow of approximately 55 l min$^{-1}$ is led to a manifold, located at the marine station, which supplies the water to different
instruments for the analysis. Sensors, their flowcaps and tubings connecting them are automatically cleaned once a day using detergent (Triton X-100), supplemented with a weekly to semiweekly manual cleaning of sensors.

**Temperature and salinity** were observed using a thermosalinograph (SBE45 MicroTSG, Seabird Scientific) using a logging interval of 15 s, connected to the flow-through system on the marine station. This measurement was supported by regular CTD
(Conductivity-Temperature-Depth) sampling close to the sample water inlet. Also, a stand-alone temperature-depth sensor (Star-Oddi Tilt and Compass), attached directly to the inlet, measured the in-situ temperature, logging data every 10–15 s. These additional observations were used for correcting the temperatures measured with the thermosalinograph, which were affected by the heat exchange between the seawater inside the pipe and the surroundings of the pipe, depending on the temperature structure of the water column.

**The partial pressure of carbon dioxide ($p$CO$_2$)** was measured using a SuperCO$_2$ system (Sunburst Sensors) that was connected to the flow-through system (Honkanen et al., 2018). The CO$_2$ of the sample water stream was equilibrated with the sample air in a double shower-head equilibrator chamber. This sample air was directed to a non-dispersive infrared gas analyzer (LI-840A, LI-COR) for the detection of CO$_2$ molar fraction (logging 15 s), which was used for the calculation of $p$CO$_2$, following the guidelines of Dickson et al. (2007). The sample water may have exchanged the heat with its surroundings
during the transport, and thus, the $p$CO$_2$ data were corrected for the temperature effect using the $T$-$p$CO$_2$ dependence given by Takahashi et al. (1993).

The drift error of the gas analyzer was compensated using a correction function, which was based on the measurement of four reference gases with differing CO$_2$ molar fractions (0, 234, 397, and 993 ppm CO$_2$) every four hours. These gases were







**Figure 2.** Structure of the flow-through system of the marine station. A submersible pump (A) transports sample water to the station (B), where the manifold distributes the water to different sensors, such as the thermosalinograh, TSG (D). The flowrates (F) to the sensors are controlled to be approximately constant. The flux tower (C) is found next to the marine station.

verified using a cavity ring-down spectroscopy analyzer (Picarro G2401), which was calibrated against gases from the National
Oceanic and Atmospheric Administration (NOAA).

A similarly operating SuperCO$_2$ system was tested in the 1st ICOS Ocean Thematic Centre $p$CO$_2$ instrument intercomparison workshop. The root mean square error between the SuperCO$_2$ and the reference setup was estimated to be 4.4 µatm (data not published). The effect of the long inlet tube on the $p$CO$_2$ measurement at Utö has also been verified to be small (Honkanen et al., 2021).

The flow-through system was offline during the spring 2017 (24 March – 31 March), early summer 2018 (1 May – 2 July) and summer 2020 (15 July – 30 July) due to the pump failure, in addition to the shorter maintenance periods. The infrared source of the gas analyzer of the $p$CO$_2$ system was malfunctioning during the spring 2020 (31 March – 26 May).





**Chlorophyll-a** was continuously measured with a Wetlabs ECO FLNTU fluorometer (Sea-Bird Scientific), logging data every 15 s, which was connected to the flow-through system. During the study period, we took discrete water samples for
laboratory analysis of Chlorophyll-a, following sampling, extraction and measuring protocols given in HELCOM (2016). Chlorophyll-a fluorescence was adjusted to represent Chlorophyll-a concentrations ($\mu$g l$^{-1}$) using the linear relationship observed between fluorescence intensity and measured concentrations ($R^2$ = 0.76, n = 159).

**Inorganic nutrients** were analyzed from the bottle samples collected from the flow-through system and frozen at -20 °C until analysis. These samples were typically collected at least monthly, and during specific measurement campaigns multiple
times per a day. The nitrate, nitrite, phosphate and silicate concentrations were analysed using a Lachat QuickChem 8000 flow injection analyser (until October 2020) and with Skalar SAN++ continuous flow analyser (from October 2020 onward) following Grasshoff et al. (1999). In this study, we were mainly concerned with oxidized forms of nitrogen (as a sum of nitrate and nitrite, referred in the text simply as nitrate) and phosphate concentrations.

### 2.3 Sea-air CO$_2$ flux

**The exchange of CO$_2$ between the sea and the atmosphere** was measured using the eddy covariance (EC) method, using a 9 m tall micrometeorological flux tower (12 m.a.s.l) on the western shore of the island. The setup consists of a closed-path non-dispersive infrared gas analyzer (LI-7000, LI-COR), connected to a 30 cm long sample air drier (PD-100T-12-MKA, Perma Pure). The EC fluxes were calculated for each 30 minute period. The CO$_2$ molar fractions observed with flux tower were regularly calibrated with the zero (0 ppm CO$_2$) and span (396 ppm CO$_2$) reference gases, in addition to the atmospheric ICOS
observations at Utö (Kilkki et al., 2015). The measured EC fluxes were corrected for the flux loss occurring in the system. For this purpose cospectrum between the vertical wind speed and CO$_2$ molar fractions was compared with the cospectrum between the vertical wind speed and the (sonic) temperature, in order to produce a frequency dependent transfer function. More information about this method is given in Honkanen et al. (2018).

The EC method was supported by flux gap-filling for those periods, when the EC data were not available due to flux
footprint originating from the island (outside of the 190–350° wind sector), non-stationary flux conditions or CO$_2$ molar fraction disturbance by ships passing the flux footprint area. The EC data were considered appropriate for 26% of the time and the remaining periods were gap-filled as described below. Based on the categorization by Rutgersson et al. (2020), the wind sector 180–260° represents open sea footprint area, while the wave field of the sector 260–350° may be disturbed slightly by distant islands (Honkanen et al., 2018). The EC fluxes during non-stationary conditions were discarded if the 30 min flux
differed more than 60 % from the mean of the corresponding 5 min fluxes. Since there exists a seaway within the flux footprint area, we applied an empirical variance restriction for the EC data to avoid interference from ship traffic. We did not use the EC data, if the variance of the CO$_2$ molar fractions exceeded 0.50 ppm$^2$ during the flux calculation period.

**A flux parameterization** based on wind speed and $p$CO$_2$ measurement was used for the gap-filling of the EC data during the periods when the EC fluxes were not available due to strict quality control. The gas transfer velocity was parameterised using
a combination of a linear and quadratic relationship of 10 m potential wind speed, $U_{10}$, which was measured at the weather station on the island, instead of using solely quadratic fits, that are commonly used (Wanninkhof, 1992, 2014). The relationship





used for the normalized gas transfer velocity, $k_{660}$ (the $k$ which is corrected to the Schmidt number in 20 °C and 35 PSU), in this paper ($k_{660} = 0.161U_{10} + 0.12U_{10}^2$) was determined for Utö based on the data from 2017–2021, by fitting the function to the 2 m s$^{-1}$ wind speed median bins. This parametrization is a site specific, and slightly differs from the one given for the open ocean conditions, $k_{660} = 0.251U_{10}^2$ (Wanninkhof, 2014). The effect of our choice on the budgets and other information on the flux measurements are given in the appendix A.

To fill in the measurement gaps in $p$CO$_2$ and in the air–sea CO$_2$ flux data, we reconstructed the $p$CO$_2$ time series during the gaps using the flux equation (Eq. 1) inversely, i.e. based on the directly measured eddy covariance fluxes and the wind-speed parametrization of gas transfer velocity. As this method may produce artificial scatter, these reconstructed $p$CO$_2$ values were averaged with a seven day running arithmetic mean filter. The $p$CO$_2$ data were available 85% of time, and thus the reconstructed and interpolated data were used for 15% of the time.

## 2.4 Other measurements

**Vertical structure of the water column** was assessed using the RBR XR-620 CTD (Conductivity-Temperature-Density) castings from a small boat northwest of Utö, where the water depth reaches 90 m. These measurements were done approximately every 10 days during summer, and less frequently in winter. The depth of the strongest temperature gradient (thermocline) was determined from each cast used for the interpretation of the mixed layer depth. If the highest density gradient was less than 0.1 °C m$^{-1}$, the water column was considered to be fully mixed.

**Meteorological observations** including wind speed and direction and air temperature were measured at Utö's Atmosphere research station, located in the eastern part of the island. The wind measurement took place at approximately 25 m.a.s.l. on relatively flat rock and bush terrain. In this study, 10 m potential wind speeds, calculated assuming the logarithmic wind profile, were used (Aaltonen, 2021). The solar irradiation was measured using a pyranometer (Delta-T SPN1) placed on the roof of a hotel in the northern part of the island, next to the ICOS Atmosphere station.

## 3 Results

### 3.1 Solar irradiation

Solar irradiation is governed by the annual cycle and the cloudiness. Generally, solar irradiance increased from almost negligible in January to the maximum in June, with a solar irradiance monthly sum of ca. 0.7 GJ m$^{-2}$ (Fig. 3). The annual solar irradiance sums of 2017 and 2018 were low, 3.54 and 3.69 GJ m$^{-2}$ respectively, compared to the other measured years with 3.86–4.08 GJ m$^{-2}$. Especially in spring–summer of 2017 and 2018, solar irradiation was low compared to other years, with the exception of May 2018. The spring and summer months had plenty of light in 2019–2021, except for May and August 2021.





## 3.2 Sea surface temperature

At Utö, the seawater temperature at approximately 4.5 m depth followed the annual cycle of solar irradiance (Fig. 3), with a delay. The temperature minima, on average 1.3 °C, occurred in February–March, whereas the temperature maxima, on average 19.7 °C, were reached in July–August. Typically around September, the water column got fully mixed, with a rapid cooling of surface water.

During the study period, the coolest summer (June–August) surface layer was observed in 2017, when the annual maximum temperature reached only 18.4 °C in late July. The temperature remained up to approximately 2 °C below the 5-year average for the most part of the year, with the exception of September–October 2017, when the cooling was slightly lower than for other years.

The year of 2018 was the warmest amongst the studied ones, in terms of the summer peak temperatures. The early spring (March–April) of 2018 was, however, still relatively cold and there was occasional on-off sea ice in March, with highly variable ice coverage. In July, the sea heated quickly from a relatively low temperature, 13 °C, to the highest temperature observed during the study period, 25 °C. The sea cooled quickly in September 2018, and the surface temperature in October was 2 °C below the average.

The temperature of the surface layer in 2019 followed closely the 5-year average. Throughout the year 2019, the monthly seawater temperatures differed less than 1 °C from the average. The annual minimum of 1.5 °C was observed in early March and the maximum of 21.3 °C in late July.

The year 2020 started particularly warm, the surface seawater remaining above 2.9 °C throughout January–March 2020. The summer of 2020 was characterized by several strong and quick temperature fluctuations. A quick early increase in temperature to 20.3 °C was observed in June 2020, but soon the temperature quickly fell to 13.6 °C. The annual maximum of 20.6 °C was observed relatively late, in late August.

In February 2021, there was varying thin on-off sea ice during two weeks. The year 2021 had the second warmest summer, and the temperature increased up to 23.5 °C in mid-July. This maximum took a place a half month earlier than the one in 2018. After late summer 2021, the sea surface remained mostly colder than the 5-year average.

## 3.3 Temperature vertical profile

The thermocline, which was determined from semi-regular CTD casts, is used here to represent the depth of the mixing surface layer, which is separated from the underlying deep water. Typically, a distinct thermocline started to develop in the spring as a result of increased solar irradiance (Fig. 4), that heated the surface waters. Typically, a very shallow (2–5 m) and short-term mixed layer formed in May. The infequent CTD casts do not reveal the exact the duration of such shallow layers. Most likely, mixed layer depths shallower than 5 m lasted only a short period of time, as these were not observed in two subsequent CTD casts, which were done in summer approximately every 10 days. In May 2018 and May 2021, strong temperature gradients of old winter layers were observed, at 37 and 57 m, respectively. The surface layers gradually deepened due to the wind induced mixing as the summer progressed, and the mixed layer typically reached the depths of 15–20 m. The summer thermocline of





2020 stayed relatively shallow from the late spring to late summer (May–August), and it never reached below 20 m, whereas the summer thermocline depth of 2017 peaked at 35 m in the beginning of June and at 32 m in the beginning of July. In 2017,

the mixed surface layer reached to 50–80 m already in September-October, in contrast to other years, and thus relatively warm waters, approximately 12 °C, reached such deep layers. Eventually, the overturning due to the water reaching the maximum density temperature, caused the water column to be fully mixed in early winter.

### 3.4 Salinity

Salinity is a good indicator for the change of water masses; for instance in the pelagic conditions of the Baltic Sea, salinity

correlates well with total alkalinity (Müller et al., 2016). In the coastal regions, salinity shifts may indicate riverine impact and upwelling. The salinity (measured at the depth of approximately. 4.5 m) at Utö typically reached its minimum in late summer, 6.1 on average (Fig. 5). After that, salinity steadily increased to the maximum met in the turn of the year, 6.8 on average. In late winter or early spring of some of the years, the salinity showed large interannual variations, and relatively large drops were observed in February 2021, as salinity fell to 5.2, and in March 2018 to 5.7.

### 245 3.5 Chlorophyll-a fluorescence

The phytoplankton biomass at Utö, inferred from the Chlorophyll-A (Chl-A) fluorescence data validated with laboratory measurements, showed seasonal variability with a range from 0.5 to 10 µg l$^{-1}$ (Fig. 6). Spring bloom formed in April, due to increased irradiance levels and water column stratification and fuelled by inorganic nutrients accumulated during winter. In 2017 spring bloom peaked very early, before mid-April, although we had some gaps in measurements. The spring bloom

of 2018 stands out from the 5-year average: the Chl-A fluorescence of April 2018 was twice as high as the 5-year average for April. The spring bloom in 2020 was very weak and delayed until May. Spring blooms typically declined in May, with the exception of 2019, which showed a secondary peak of Chl-A after mid-May. Additional summer blooms were seen in July–August.

### 3.6 Nutrient concentrations

The seasonal dynamics of nutrient concentrations were controlled by the production, remineralization and mixing. The levels of nitrate and phosphate at the depth of 4.5 m reached their annual maxima in February–March, before the spring bloom (Fig. 7). For both of these inorganic nutrients, the highest pre-spring levels were observed in 2018 and 2021, approximately 8–10 µmol l$^{-1}$ for nitrate, while the pre-spring nutrient levels for the year 2020 were low, less than 6 µmol l$^{-1}$ for nitrate, compared to the 5-year average. The pre-spring nutrient data for Utö in the year 2017 were not available. However, based on the early

April 2017 data, the nutrient inventories had been depleted earlier than in the other years.

The nitrate levels at Utö reached the minimum (0.5 µmol l$^{-1}$) in May and predominantly stayed this low until September. However, the nitrate levels in 2020 increased to 1.7 µmol l$^{-1}$ in June–July. In autumn, all years showed a similar increase in the nitrate.




Similarly to nitrate, the phosphate concentrations in July 2020 deviated from other years as the monthly average increased
to 0.5 µmol l$^{-1}$. The phosphate levels reached the minima in August with the monthly average of 0.2 µmol l$^{-1}$. The lowest
concentrations were observed during July–August 2021. In autumn, the phosphate inventories filled up.

### 3.7   Wind speed

Wind is an important factor enhancing the turbulence in the air–sea interface and thus it plays an important role in the deepening
of the surface layer and in the gas transfer between the sea and the atmosphere. The highest wind speeds occurred during the
wintertime, October–February, monthly averages being 8.0 m s$^{-1}$, and the lowest monthly wind speeds, on average 5.3 m s$^{-1}$,
took place in summer, May–June. The late winter of 2019–2020 was relatively windy, as the monthly averages for January and
February were 9.6 m s$^{-1}$. The lowest monthly mean of wind speed (4.5 m s$^{-1}$) was observed in May 2021.

### 3.8   $CO_2$ partial pressure

The partial pressure of $CO_2$ in the seawater (Fig. 9) generally followed the annual cycle primarily governed by the photo-
synthetic carbon fixation. In spring, when the solar irradiance increased and water column started to be stratified, increased
phytoplankton production transformed the inorganic carbon into organic carbon, resulting in a drawdown of the $p$CO$_2$. On av-
erage, the $p$CO$_2$ reached the minimum of 210 µatm in late April. The spring bloom was typically terminated by the depletion
of inorganic nitrogen (Fig. 7). In 2017, $p$CO$_2$ drawdown started approximately a week earlier than in other years, and also the
peak of spring bloom took place earlier. The lowest spring $p$CO$_2$ was observed in 2018 and 2019, concomitantly with the most
intense spring blooms.

The $p$CO$_2$ generally increased from late April (210 µatm) to early July (340 µatm), after which it experienced another
drawdown to 220 µatm. This late summer drawdown showed some temporal variation, taking place from late June to late July.
The lowest half-hourly $p$CO$_2$ value (58 µatm) recorded during 2017–2021 occurred in mid-July 2021. During the summer
months of 2017, the daily averages of $p$CO$_2$ remained relatively high compared to other years, not falling below 220 µatm.

In late summer, from August onwards, the solar irradiance started to diminish, decreasing the photosynthesis rate, and the
respiration of organic matter started to prevail, which was observed as increasing $p$CO$_2$. This effect was enhanced by the
breakdown of the stratification of the water column. In August–September, $p$CO$_2$ generally increased almost linearly to 630
µatm, where the annual maxima were met. In September 2019, $p$CO$_2$ reached the highest recorded $p$CO$_2$ during the study
period, 877 µatm (half-hourly value). After this, $p$CO$_2$ decreased slowly until the spring, approaching the atmospheric $CO_2$
partial pressure.

### 3.9   Air–sea $CO_2$ flux

The $CO_2$ fluxes between the sea and the atmosphere at Utö (Fig. 9) followed an annual cycle governed by the $p$CO$_2$. During
the spring—summer (approximately April–August) when the seawater $p$CO$_2$ was below the atmospheric partial pressure of
$CO_2$, the sea was a sink of atmospheric carbon dioxide. The sea was a source of $CO_2$ during the time between the autumn





| Year | $CO_2$ balance, best fit | $CO_2$ balance, Wanninkhof (2014) | Sink | Source | Uncertainty |
|------|------|------|------|------|------|
| 2017 | 39.1 | 37.2 | -21.4 | 60.4 | 17.9 |
| 2018 | 18.2 | 19.9 | -30.6 | 48.8 | 7.9 |
| 2019 | 23.2 | 22.3 | -37.2 | 60.5 | 10.1 |
| 2020 | 28.2 | 27.2 | -25.8 | 54.0 | 11.3 |
| 2021 | 26.5 | 26.2 | -27.4 | 53.9 | 13.7 |

**Table 1.** The $CO_2$ balance calculated using the gap filling with the best fit of the gas transfer velocity, the balance using the parametrization by Wanninkhof (2014), sum of the sink fluxes, sum of the source fluxes and the uncertainty of the annual air-sea $CO_2$ flux at Utö in 2017–2021. The positive values indicate annual $CO_2$ emissions. The unit of the values is gC m$^{-2}$ y$^{-1}$.

and the spring. The highest average negative (from the atmosphere to the sea) monthly flux (-0.3 μmol m$^{-2}$ s$^{-1}$) occurred in May, which coincidenced with the lowest average monthly $pCO_2$. In May–July 2017, the monthly fluxes remained clearly below the corresponding 5-year averages. The highest positive fluxes (0.4 μmol m$^{-2}$ s$^{-1}$) occurred in October, which was also the month having the highest $pCO_2$ values. In winter, the wind speeds were typically higher than in summer, enhancing the gas transfer. As the $CO_2$ partial pressure difference between the sea and atmosphere decreased in winter, the air–sea $CO_2$ flux

decreased simultaneously, slowing the $pCO_2$ decrease. Most (95 %) of the instantaneous half-hourly air–sea $CO_2$ fluxes were within -0.56 – 0.85 μmol m$^{-2}$ s$^{-1}$.

Based on the air–sea $CO_2$ flux data collected within 2017–2021 at Utö, the sea was a source of carbon dioxide (Table 1). On average, 27.1 grams of carbon was released for each square metre annually. The highest net source strength (39.2 gC m$^{-2}$ y$^{-1}$) was observed in 2017, whereas the following year, 2018, showed the lowest emission (18.2 gC m$^{-2}$ y$^{-1}$).

The cumulative sums of the air–sea flux $CO_2$ in 2018 were low throughout the year, compared to the 5-year average (Fig. 10). The initial $pCO_2$ in January 2018 was low, generating only small sea-to-air fluxes in the beginning of the year. The spring bloom in 2018 was strong, which made the cumulative sum of the air–sea $CO_2$ flux stay low over the summer. The increase of the monthly flux sums in the latter half of the year was relatively ordinary.

For the year 2017, the high initial $pCO_2$ in January caused the year to start with a high cumulative $CO_2$ emission. The bloom

was weak and the $pCO_2$ stayed relatively high for the summer, causing the cumulative sum to raise above the average sum. Based on the monthly sum slopes, the release of $CO_2$ was ordinary during September–December. The difference to the average had already occurred before autumn–winter, and it remained this way for the rest of the year. Interestingly, the cumulative sums in Septembers 2017 and 2020 were similar, but the $CO_2$ release within October–December 2020 was low.

The annual air–sea $CO_2$ flux budgets calculated for the calendar year (1 January – 31 December) are affected by the impact

of the previous year or how quickly the sea releases $CO_2$ to the atmosphere in autumn. For this reason, it would be more scientifically relevant to study the sink and source seasons separately. For this purpose, we calculated the sum of the air–sea $CO_2$ sink (negative) fluxes for each year (Table 1). The average annual sum of sink $CO_2$ fluxes was -28.5 gC m$^{-2}$ y$^{-1}$, with a standard deviation of 5.9 gC m$^{-2}$ y$^{-1}$ and coefficient of variation of 21% (standard deviation divided by the average). The



average sum of source (positive) fluxes was 55.5 gC m$^{-2}$ y$^{-1}$, and the standard deviation of it was 5.0 gC m$^{-2}$ y$^{-1}$ or 9 %.
The coefficient of variation suggests that the summer $CO_2$ sink fluxes vary relatively more than the annual source fluxes.

## 4 Discussion

### 4.1 Net air–sea $CO_2$ exchange

The marine ecosystem at Utö acted as a source of atmospheric carbon dioxide, on an annual basis in 2017—2021. The average air–sea exchange of $CO_2$ to the atmosphere at Utö was 27.1 gC m$^{-2}$ y$^{-1}$, which is comparable to the $CO_2$ balance measured
in pelagic conditions in the Gulf of Bothnia (Algesten et al., 2006). Also, the $pCO_2$ variation at Utö, approximately 200–600 µatm, is similar to the one observed in the Gulf of Bothnia (Algesten et al., 2004). In the Baltic Proper, for comparison, the $pCO_2$ maximum in autumn–winter is lower, appoximately 500 µatm (Schneider et al., 2014). Being approximately 80 km from the mainland, the marine ecosystem at Utö is coastal but it still differs clearly from the European estuaries where $pCO_2$ can peak up to almost 10000 µatm and instantaneous fluxes range within 1–9 µmol m$^{-2}$ s$^{-1}$ (Frankignoulle et al., 1998).
The net flux of $CO_2$ to the atmosphere suggests that the sea surface system receives carbon (organic or inorganic) from elsewhere (Cole et al., 1994). Organic carbon originates primarily from photosynthesis, either locally or imported from elsewhere. Microbial degradation and sun-induced photodegradation of organic carbon remineralizes carbon and nutrients in the seawater into inorganic forms. Land-aquatic continuum systems are typically strong $CO_2$ sources due to the net heterotrophy (respiration overweighing the production), stimulated by the high river inputs (Borges et al., 2005). For instance, the Gulf of Bothnia is a
net source of $CO_2$ to the atmosphere due to the low productivity and high input of terrestrial organic matter (Algesten et al., 2006; Kuliński and Pempkowiak, 2011).

Phytoplankton blooms dictate the quantity of inorganic carbon fixation ($pCO_2$ drawdown). A part of this biologically fixed carbon is respired and mixed back to the sea surface system eventually, and another part is removed from the short term carbon cycle due to e.g. sedimentation, refractory compounds or grazing. Mixing events bring $CO_2$ rich water back to the surface
layer. Such an upwelling event probably occurred in July 2020, when simultaneous drops in surface layer temperature, nutrient concentrations and Chl-A concentration and increases in salinity and the $pCO_2$ were observed. In winter, the $pCO_2$ is mainly governed by the outgassing of the inorganic carbon into the atmosphere, as it slowly approaches the atmospheric $pCO_2$. The carbon balance of the system is also governed by advection, which can depend on e.g. current patterns and river dynamics, and the atmospheric deposition. Undoubtedly, the magnitude of many of these carbon flows remain unknown, without a large set
of multidisciplinary measurements.

The possible advection of carbon to the sea surface system at Utö might explain some differences between the observed $pCO_2$ variation and the $pCO_2$ variation calculated from the oxygen using the Redfield ratios in the study of Honkanen et al. (2021), especially concerning the seasonal variability. Southward water transport in the Archipelago Sea is typically largest in spring–summer (Miettunen et al., 2023), bringing riverine water to Utö which could increase the observed $pCO_2$ compared to
the $pCO_2$ calculated from the oxygen changes assuming Redfield ratios between the carbon and oxygen.





## 4.2 Interannual variability of the net air–sea $CO_2$ exchange

The annual net air–sea $CO_2$ exchanges at Utö in 2017–2021 varied within 18.2–39.1 gC m$^{-2}$ s$^{-1}$. The magnitude of this interannual variability is higher than the one observed in the pelagic conditions in the Baltic Proper, -7 – -11 gC m$^{-2}$ s$^{-1}$ (Schneider et al., 2014). Two years differing the most from the mean of this period (2017 and 2018) are discussed here in 355 detail. Especially, the summers of these two years exhibited notable differences in the $pCO_2$ and air–sea $CO_2$ flux dynamics.

The highest annual balance of $CO_2$ (39.1 gC m$^{-2}$ s$^{-1}$) occurred in 2017. The $pCO_2$ in 2017 fell relatively early, already in late March. The nitrate concentration was depleted already in April, and there were no more nutrients to fuel further the spring bloom. The algal biomass quickly decreased from April to May. The seawater $pCO_2$ in the summer 2017 remained extraordinarily high, compared to other years. Therefore, the $pCO_2$ gradient between the sea and atmosphere remained low 360 over the summer, and the sink fluxes were accordingly lower than in other years. The summer of 2017 was also the coolest of the five study years. The conditions did not favour phytoplankton production due to relatively low amount of sunlight and nutrients, and the occasionally deep mixing depth. The thermocline depth was deep in the late summer of 2017, affecting the surface temperatures. This mixing of $CO_2$ rich water to the surface may have diluted the drawdown of $pCO_2$ at the surface, thus decreasing the negative fluxes in summer. The source fluxes in the early winter 2017 were high, even though the $pCO_2$ 365 gradient was ordinary or even small, compared to other years, but these months were very windy, which increased the gas transfer velocity. Due to this, the sea was able to release more carbon dioxide to the atmosphere before the end of the year.

The year 2018 had the lowest net $CO_2$ source observed at Utö in 2017–2021. In 2018, the spring bloom, observed from Chl-a fluorescence, was high compared to other years, quickly reducing $pCO_2$ in April. In addition to the expressive spring bloom, 2018 also had the warmest summer, with the surface temperature reaching up to 25.0 °C. Temperature affects directly 370 the dissociation and solubility of $CO_2$, but the warm and calm surface water conditions also support the cyanobacterial blooms. Actually, in summer 2018, an intense cyanobacteria bloom was observed from mid-July until end of July (Kraft et al., 2021). This was partly seen in Chl-a fluorescence as well, although that method is not well suited for cyanobacteria observations.

Wesslander et al. (2010) found that the interannual variability of the summer $pCO_2$ in the central Baltic Sea and Kattegat are governed by the maximum phosphate concentrations. The highest annual sum of the sink air–sea $CO_2$ fluxes was observed 375 in 2019, and this year had an average concentrations of pre-spring nutrients. Thus, the nutrients alone do not predict the whole interannual variation of air–sea $CO_2$ exchange at Utö. However, the nutrient levels in connection with other environmental conditions play an important role governing the spring–summer blooms. Low saline conditions at Utö in February 2021 and March 2018 may be indicative of intrusion of riverine water masses, inputing the nutrient inventories. The pre-spring nitrate and phosphate levels were high in both 2018 and 2021.. The spring bloom of 2021 was ordinary at most, in terms of $pCO_2$, 380 possibly due to the low solar irradiation levels in May 2021. However, the late summer bloom in July 2021 generated record low $pCO_2$ and phosphate levels. The late summer increase of phytoplankton was possibly related to the blooms of cyanobacteria, which has been observed at Utö using imaging flow cytometry (Kraft et al., 2022).



The initial $pCO_2$ in January had some impact on the annual air–sea $CO_2$ exchange as seen as the difference between Januaries of 2017 and 2021. On average, the $pCO_2$ was approximately 500 µatm in the beginning of the year. The year 2017 started with

a very high $pCO_2$ (600 µatm), which partly caused the $CO_2$ fluxes to be higher during January than for the other years.

If the inorganic carbon system at Utö is driven by the organic carbon input entering the system, we should see a difference in the annual air–sea $CO_2$ fluxes depending on the dissolved organic carbon concentrations and possibly even on the riverine fluxes of organic matter. Based on the model results by Miettunen et al. (2023), during the first half of the year 2017, the water masses advected southwards in the Archipelago Sea, possibly bringing organic rich waters to our study area. Additional

measurements directing to organic carbon are required to understand these diverse carbon flows.

### 4.3 Importance of the $CO_2$ emissions in coastal ecosystems

The coastal seas are vulnerable ecosystems, highly impacted by human actions and climate change. These areas provide humans with multiple ecosystem services, ranging from fisheries to carbon sequestration. River discharges into the Baltic Sea are expected to increase with increasing precipitation, which would generate an increased inflow of organic matter (Asmala et

al., 2019).

This study demonstrates the importance of multidisciplinary studies combining atmospheric, marine and ecosystem research with continuous long-term observations. The results present the annual carbon balance between the sea ecosystem and the atmosphere in a coastal sea region, where the biogeochemical processes dominate the annual cycle, in contrast with the open oceans, where the main factor impacting the carbon balance is temperature. The coastal environments, such as marginal seas,

upwelling systems and estuaries, exhibit one of the largest flows of oceanic carbon and nutrients (Borges et al., 2005).

As the carbon uptake by the marine ecosystem forms the base for marine ecosystem food chain and the ecosystem diversity is largest in coastal areas, these kinds of observations are necessary both for climate change and global biodiversity change studies. These results showing the $CO_2$ emissions raise questions on further need of drainage basin management, as the organic matter fluxes in the land-sea continuum affect the carbon balance of the sea. The carbon balance of the terrestrial ecosystems are

currently included in the national legislation, while the role of the marine ecosystems is omitted, even though the management of the marine ecosystems has been recognised as a way for the mitigation of climate change (Hilmi et al., 2021). Certainly, the carbon flows between the land and oceanic ecosystems require more scientific and public attention.

## 5 Conclusions

We analyzed the 5-year (2017–2021) data set of the air–sea $CO_2$ flux measurements made at Utö Atmospheric and Marine

Research Station. The sea area around the Utö island was found to be act as a net source of $CO_2$ with an average annual net air–sea $CO_2$ exchange of 27.1 gC m$^{-2}$ y$^{-1}$. The annual exchange varied between 18.2 in 2018 and 39.2 gC m$^{-2}$ y$^{-1}$ in 2017. Emission of $CO_2$ indicates that the marine system at Utö respires carbon originated elsewhere, and this possible carbon advection likely involves interannual variability, depending on many meteorological, oceanic and anthropogenic factors.



The air–sea $CO_2$ exchange at Utö underwent a large biologically driven seasonal cycle during each year, with a $CO_2$ sink
in spring–summer and a source in autumn–winter. The interannual variability in the $pCO_2$ was greatest in summer, which
suggests that the summer $pCO_2$ level had a significant effect on the interannual variability of the net air–sea $CO_2$ exchange.
The strength of the algal blooms contributed to the magnitude of the $pCO_2$ drawdown in spring and summer. High pre-
spring nutrient concentrations favoured the strong bloom during the warm summer of 2018 and thus generated a strong $pCO_2$
drawdown. During the cloudy and cold summer 2017, the $pCO_2$ stayed relatively high compared to the other summers studied.

Typically the calculation of the net air–sea $CO_2$ exchange is based on a calendar year, which may be impacted by the initial
$pCO_2$ level at the start of each year. If the autumn of the previous year had low exchange of carbon with the atmosphere, due to
the low wind speeds, the inorganic carbon inventories have remained relatively high for the turn of the year. As the exchange
is directly proportional to the $pCO_2$ gradient, this increases the fluxes in the beginning of the year.

The net exchange of $CO_2$ between the atmosphere and the sea at Utö is likely governed by the interplay between the
advection, biological processes and sedimentation. The advection replenishes the carbon inventories, while production and
sedimentation consume inorganic carbon. Thus, the interannual variability should originate from the yearly variability of these
processes. The sustained multidisciplinary research made at Utö can pinpoint the effect of these different carbon flows on the
air–sea $CO_2$ exchange in future.

The dataset used here can be found from the open archives of Zenodo.

**Appendix A: Uncertainty of the parameterized air-sea $CO_2$ fluxes**

Both the direct measurements of $CO_2$ flux by EC and the flux calculation based on the measured $pCO_2$ gradient and wind
speed introduce uncertainty in the net $CO_2$ exchange estimates.

**The EC method** measures directly the $CO_2$ exchange between the surface and the lower atmosphere within a given flux
footprint area. The sink-source-strength of the surface equals the vertical EC flux measured above the surface on a flat and
homogeneous terrain under steady state conditions. The relative nonstationarity ($RN$), introduced in Sect. 2.3, is an important
variable for analysing the fulfilment of the steady state assumption (Honkanen et al., 2018). Here, we used a relaxed $RN$
threshold (60%). We calculated the root mean square error, $RMSE$, between the measured and parameterized fluxes for
different $RN$ thresholds, and found out that the $RMSE$ decreases sharply when RN is tightened from 100% (0.45 µmol m$^{-2}$
s$^{-1}$) to 60% (0.35 µmol m$^{-2}$ s$^{-1}$), but remains stable below this threshold: at $RN = 10\%$, $RMSE = 0.33$ µmol m$^{-2}$ s$^{-1}$.

**The parameterized air–sea $CO_2$ flux** relies on the measurement of the $CO_2$ gradient at the sea-atmosphere interphase,
calculation of $CO_2$ solubility and the parametrization of the gas transfer velocity, which relates to the intensity of the turbulence
at this interphase (Eq. 1).

The measurement of the $CO_2$ gradient is a technical challenge. The pumps used for the sample water intake must typically
always be wet, and thus, the sample water inlet cannot be placed directly at the water surface. Temperature and $pCO_2$ gradients
between the inlet depth and the water surface can generate an error on the estimate of the air–sea $CO_2$ gradient, especially,
during summertime, when a shallow thermocline develops. As these layers shallower than the water inlet are relatively short-



term, their effect is not dealt with here. An assessment of the overall effect of $pCO_2$ gradients would require vertical $pCO_2$ measurements within the mixed surface layer.

Sea ice, as a physical barrier separating the sea from the atmosphere, can generate differences between the actual fluxes and parameterized air–sea $CO_2$ fluxes. We observed sea ice at Utö only occasionally in March, when the $pCO_2$ difference between the sea and the atmosphere, and thus the flux between them, is typically small. Also, a solid coverage of sea ice was present only during short periods, and thus the sea ice did not play a significant role in the processes affecting net air–sea $CO_2$ fluxes.

In order to present full timeseries of $pCO_2$, we reconstructed small part of the $pCO_2$ data with the method described in Sect. 2.3. The $R^2$ between this reconstructed and measured $pCO_2$ was 0.74 and the $RMSE$ was 69 µatm.

The most common parametrizations of the gas transfer velocity are based on wind speed (Wanninkhof, 2014), but they are best suited for the open ocean conditions, where the wind can generate waves freely. As there are numerous islands in the Archipelago Sea, we developed a local parametrization from our measurements of gas transfer velocity and wind speed.

We analyzed the measured gas transfer velocity as a function of wind speed. For this analysis, we tightened the threshold of relative non-stationarity from 60% to 30%. We also used similar data processing as done by Gutierrez-Loza et al. (2022), and included only the periods, when the absolute $CO_2$ partial pressure difference between the sea and the atmosphere was at least 50 µatm. Also, the measured negative gas transfer velocities were not used. The $RMSE$ between the parameterized and measured air–sea $CO_2$ flux for these data (n=8758) was 0.30 µmol m$^{-2}$ s$^{-1}$ or 33% error.

The normalized gas transfer velocity, $k_{660}$, parametrized according to Wanninkhof (2014) generally followed the measurements (Fig. A1) but tended to underestimate for the most common wind speeds (< 10 m s$^{-1}$) and overestimate for the highest wind speeds (> 15 m s$^{-1}$). This could have an effect especially for the negative air–sea $CO_2$ fluxes in summer, when the wind speeds are low. In low wind conditions, it is possible that the coastal charasteristics, such as cross-swell, of the study site increase the turbulence in the air–sea interface. With high wind speeds, on the other hand, the fetch may turn out to be a limiting factor for the wave growth but this applies to a shorter period of time. For comparison, we also calculated the net $CO_2$ exchange at Utö using both the Wanninkhof (2014) and the local parametrization, and found only a small difference. The average net $CO_2$ exchange, calculated using the open ocean parametrization was 26.6 gC m$^{-2}$ y$^{-1}$, and for different years the annual exchanges are given in Table 1.

The error estimates for the monthly air–sea $CO_2$ fluxes and annual balances were calculated assuming that the parametrization of the gas transfer velocity is the dominant source of uncertainty. The monthly median absolute difference between the EC flux and parameterized flux was used as an error estimate for the monthly fluxes, and this estimate was weighted by the proportion of parametrized fluxes. The error estimate for the annual net exchange of air–sea $CO_2$ was obtained by multiplying the annual balance with the median absolute percentage error (between the EC and parameterized flux), weighted by the proportion of parametrized fluxes. The uncertainty for the average net exchange of $CO_2$ was 11.3 gC m$^{-2}$ y$^{-1}$. The error estimates for each year are given in Table 1.

*Author contributions.*



SK, TM, JS, PY and LL designed and constructed the flow-through system and the micrometeorological flux tower. KS, SK, JS, LL, LH, PY and MH were charge of the operation of the flow-through measurements. TM, JH, LL and MH were operating the flux measurements. JH was in charge of the flux software. MH created the data figures and performed the data analysis. MH led the conceptualization of the research and the writing of the text, which were contributed by LH, MA, JS, JPT and LL. SMS created the figure 1 and SK created the figure 2.

*Competing interests.*

There are no competing interest between the authors.

*Acknowledgements.* We thank Ismo and Brita Willström for the CTD casts and maintenance of Utö station, and everybody else involved in Utö's measurements. We acknowledge the Integrated Carbon Observation System (ICOS) for providing the atmospheric $CO_2$ data at Utö. This study utilized the data from the Utö Atmospheric and Marine Research Station, which is part of the national Finnish Marine Research
Infrastructure (FINMARI) consortium and Joint European Research Infrastructure for Coastal Observatories (JERICO).

This research has been supported by the Research Council of Finland project SEASINK (Evolving carbon sinks and sources in coastal seas – will ecosystem response temper or aggravate climate change? project nos. 317297 and 317298). Part of this work was supported by the JERICO-NEXT and JERICO-S3 projects which have received funding from the European Union Horizon 2020 Research and Innovation Program under grant agreement nos. 654410 and 871153, respectively.

MH's work has been financially supported by a Research Council of Finland project SEASINK (grant nos. 317297 and 317298) and EU H2020 projects JERICO-NEXT and JERICO-S3 (grant nos. 654410 and 871153, respectively). LH received funding from the Research Council of Finland grant agreement No. 342223.



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



## (a) Global solar irradiation

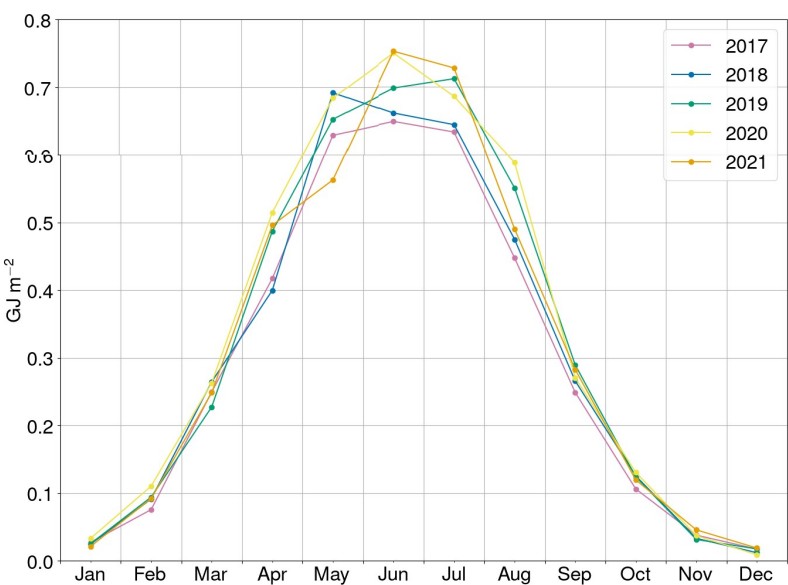

## (b) Seawater temperature (-4.5 m)

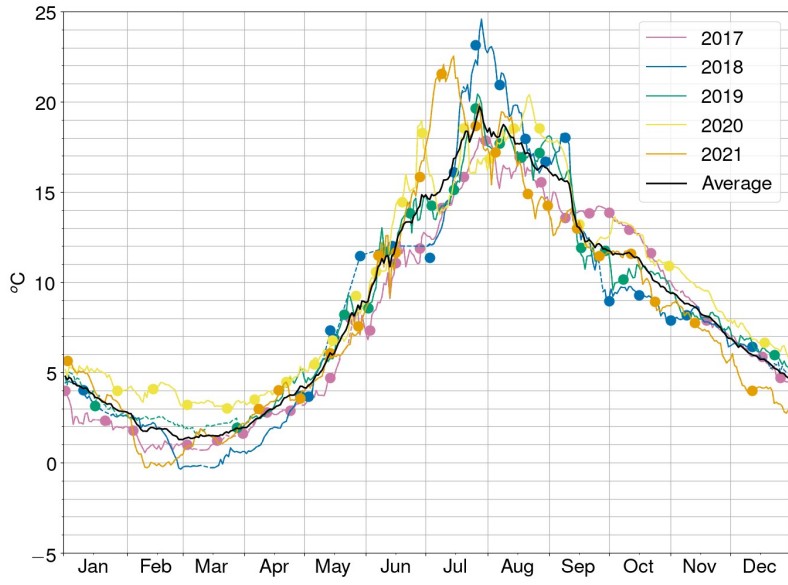

**Figure 3.** (a) Monthly sums of the global solar irradiation at Utö during 2017–2021, (b) Daily averages of the seawater temperature at the depth of approximately 4.5 m at Utö for the years 2017–2021. The solid line shows the inlet temperature measured using the standalone sensor attached to the inlet for May 2019 – May 2021, and for other time, the solid line is a thermosalinograph temperature that is shift-corrected based on the CTD measurements at 4–5 m depth. The interpolated values are indicated by the dashed lines. The CTD temperatures for the depth 4–5 m are the indicated by large dots.




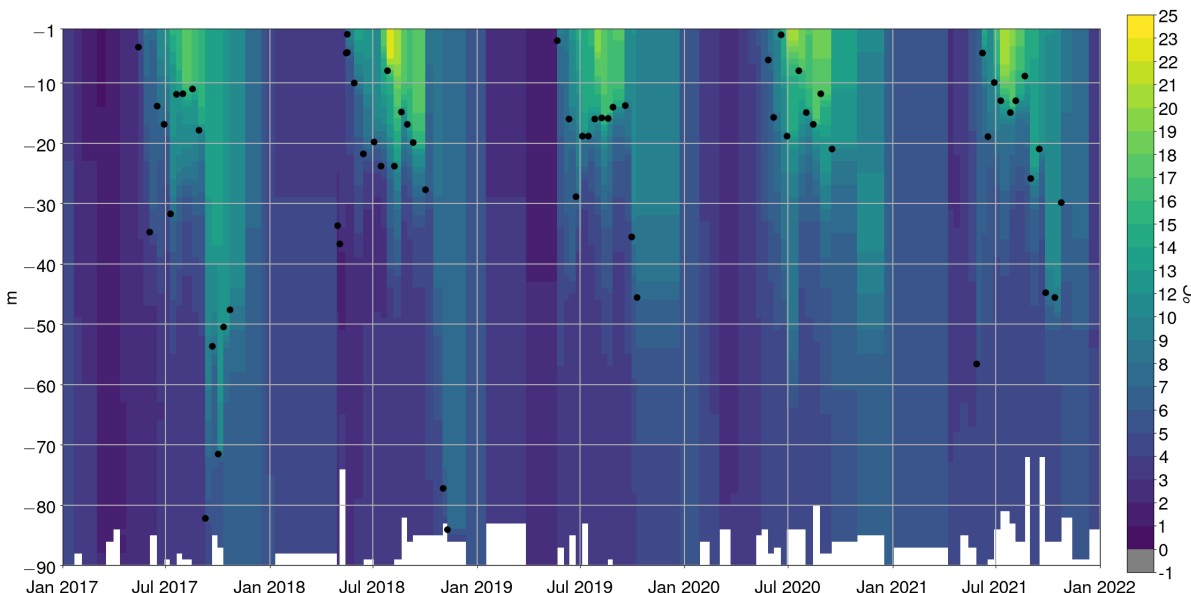

**Figure 4.** Temperature vertical profiles (colours) and the thermocline depths (black dots) at Utö for the years 2017–2021, derived from the CTD casts.



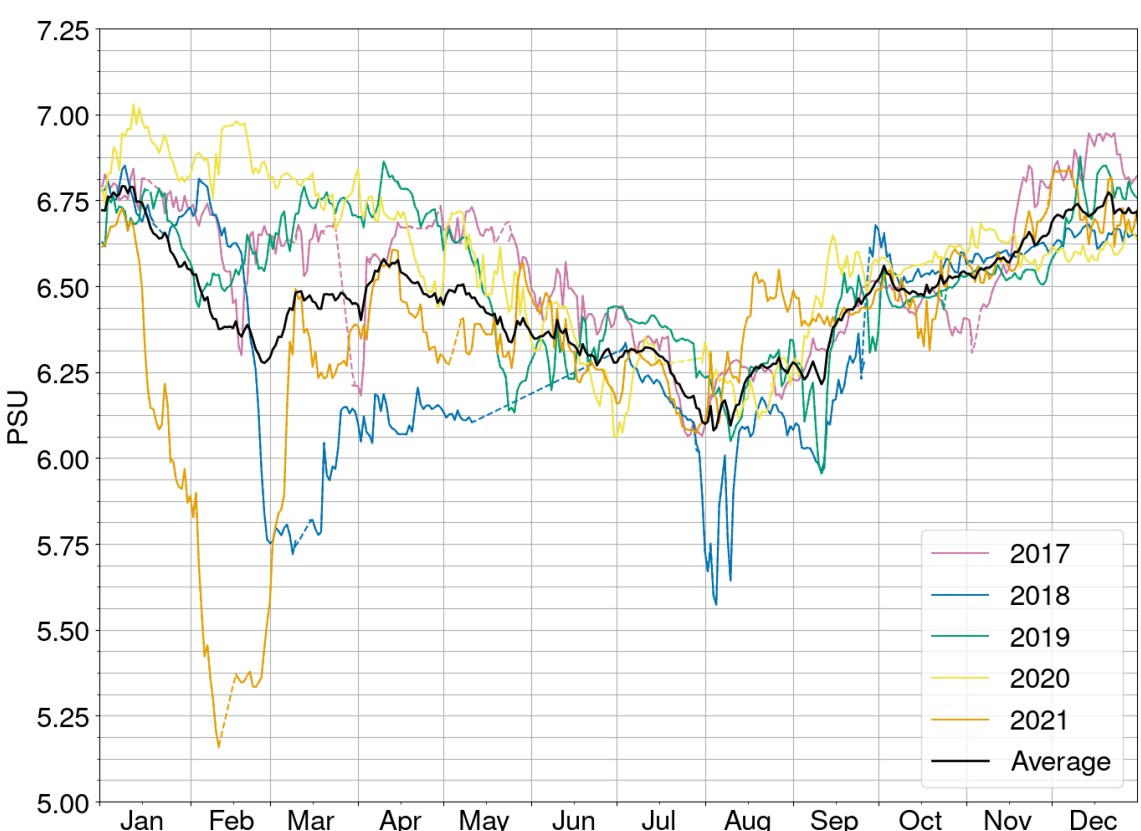

**Figure 5.** Salinity in the practical salinity units (PSU) at the depth of approximately 4.5 m Utö during 2017–2021. The interpolated values are depicted as a dotted line.





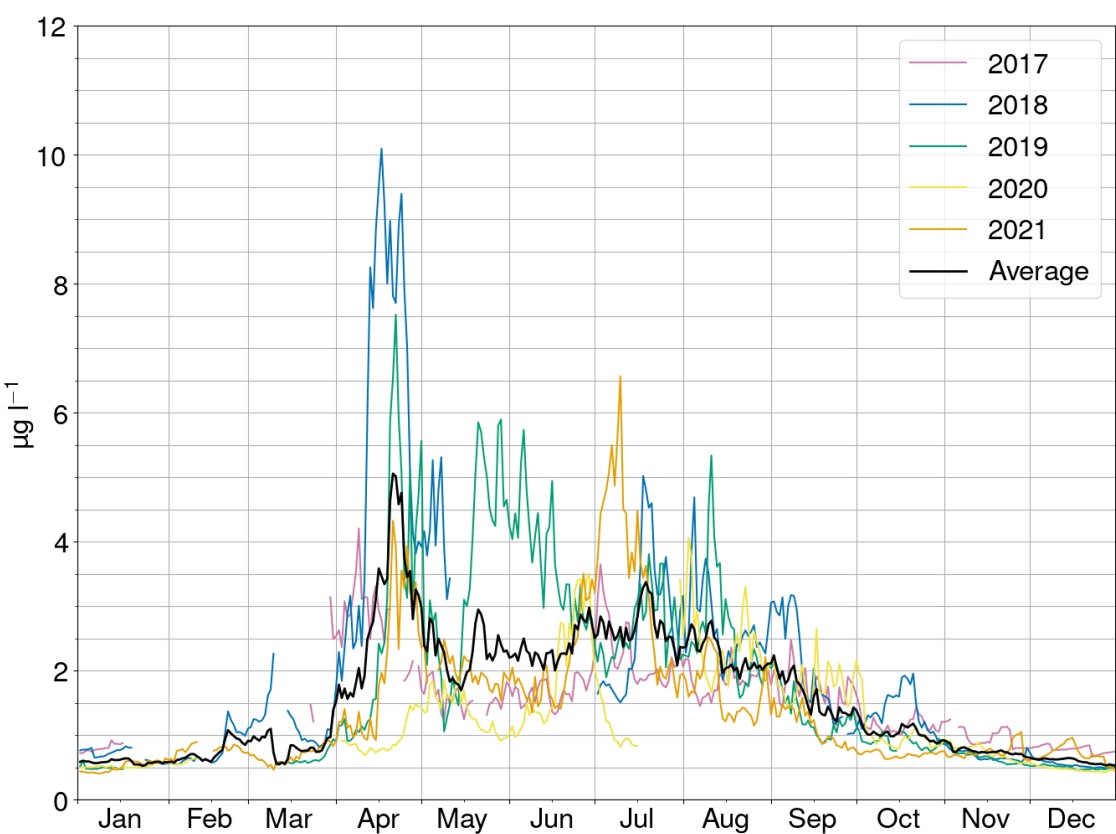

**Figure 6.** Daily means of the Chl-A concentrations at the depth of approximately 4.5 m at Utö for the years 2017–2021.



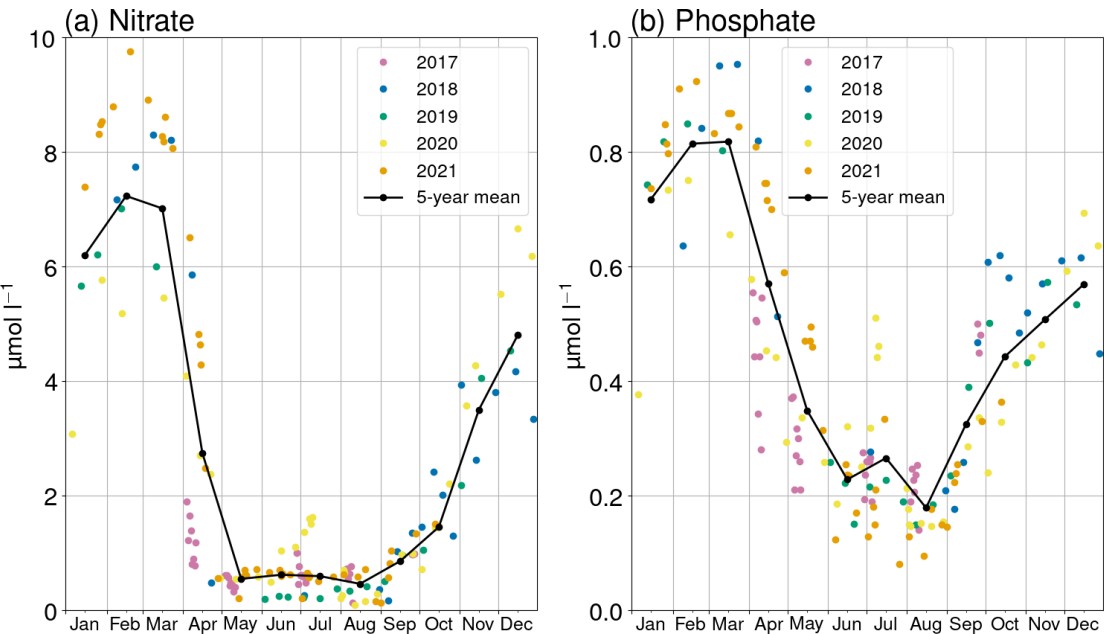

**Figure 7.** Daily averages for each year and a 5-year monthly averages, connected with lines of interpolation, of (a) the nitrate concentration and (b) phosphate concentration at the depth of approximately 4.5m at Utö for the years 2017–2021.



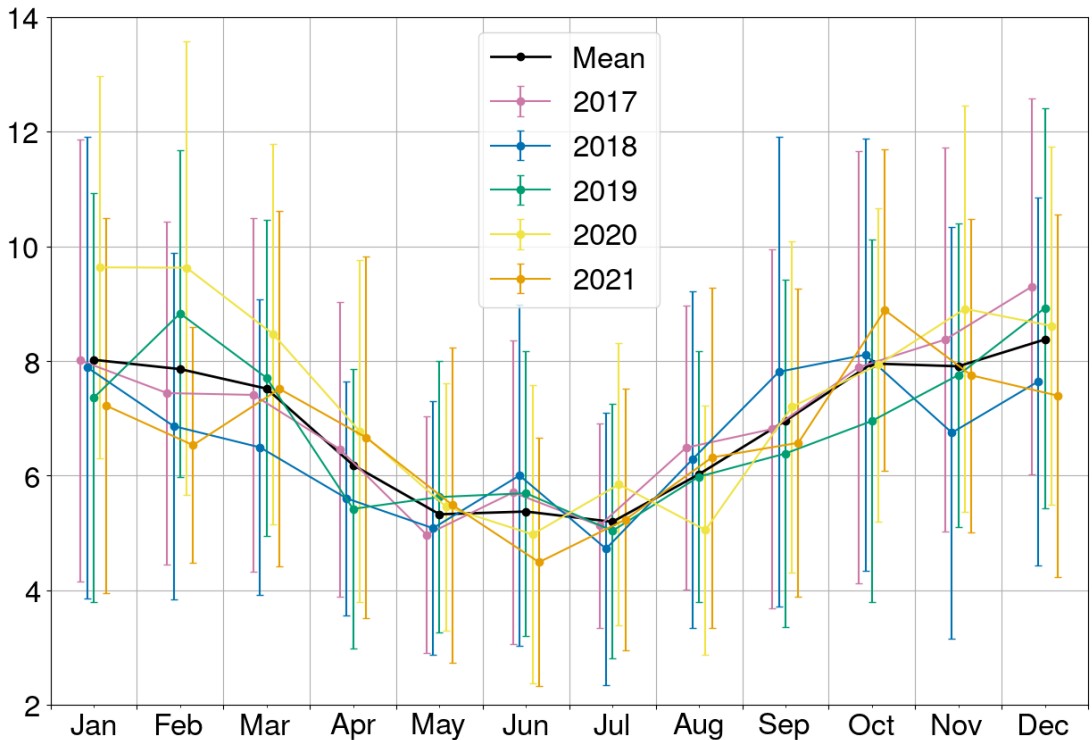

**Figure 8.** Monthly means and standard deviations of the 10 m winds peed at Utö for the years 2017–2021. Values for different years are shifted horizontally in order to increase the visibility of the different standard deviations.



## (a) Seawater carbon dioxide partial pressure (- 4.5 m)

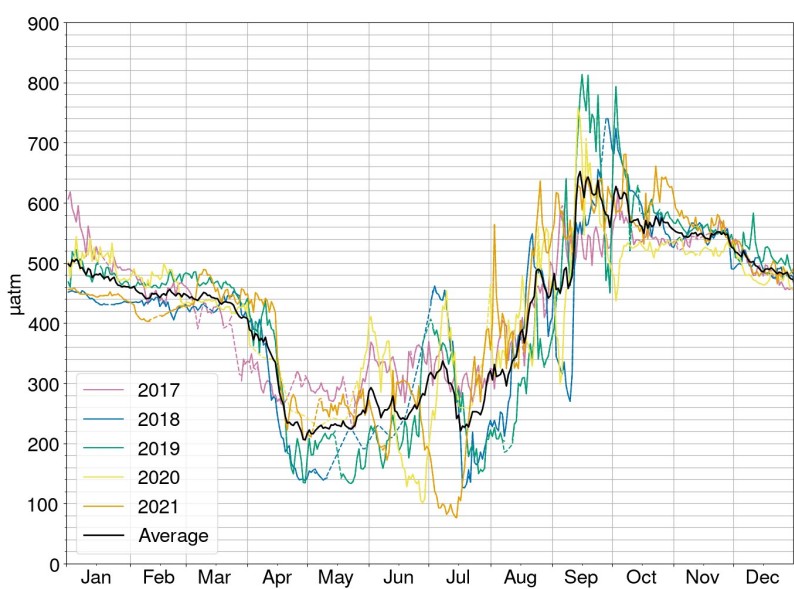

## (b) air-sea flux of carbon dioxide

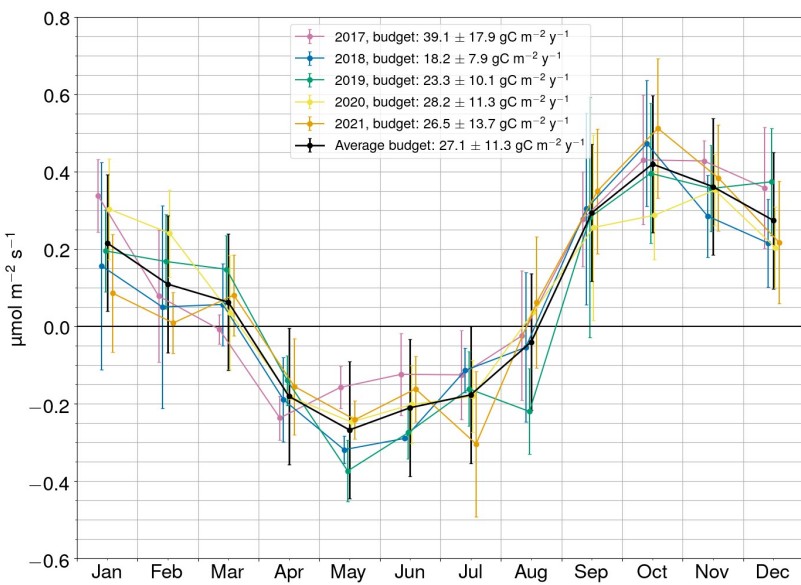

**Figure 9.** (a) Daily means of the seawater $p$CO$_2$ at the depth of 4.5 m at Utö in 2017–2021. Interpolated observations are shown with dotted line. (b) Monthly average CO$_2$ air–sea fluxes. The error estimates are explained in the Appendix.



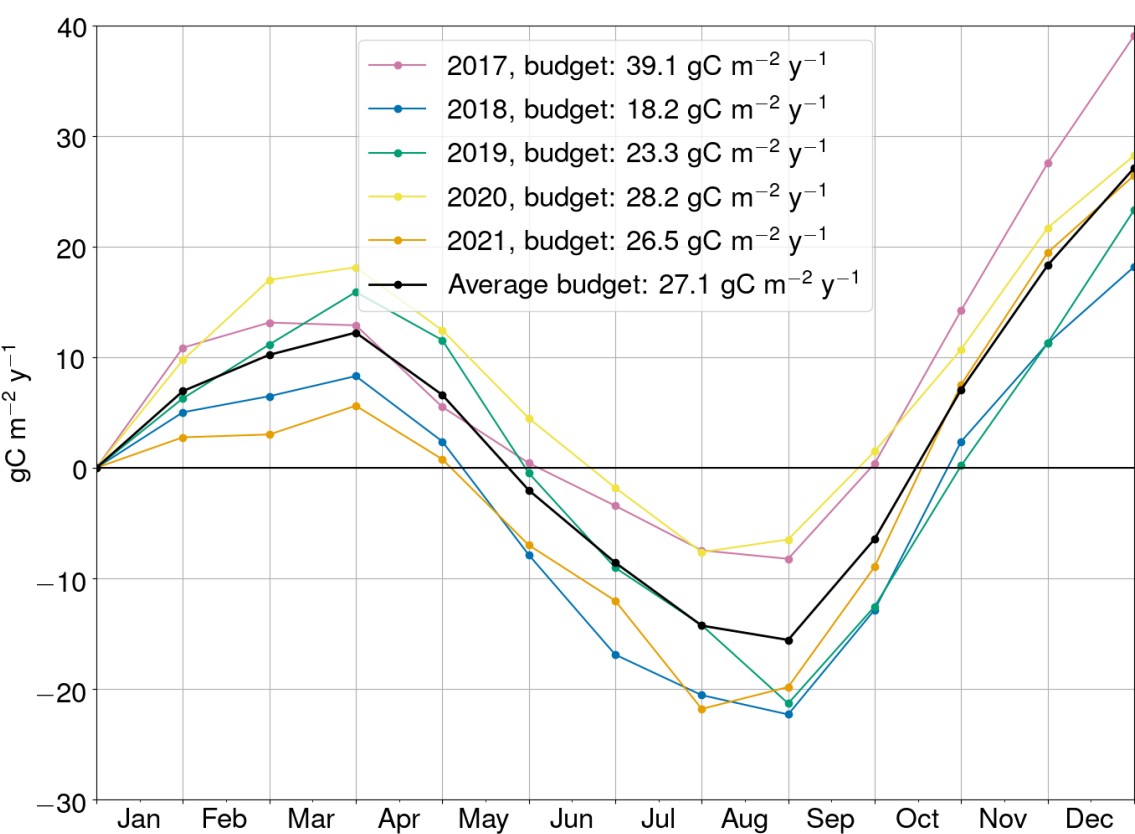

**Figure 10.** Cumulative sum of the monthly means, starting from January 1st, of the air–sea $CO_2$ fluxes at Utö for the years 2017–2021.







**Figure A1.** Gas transfer velocity, normalized to the Schmidt number of 660, as a function of wind speed.