# Peer review of "Interannual and seasonal variability of the air–sea $CO_2$ exchange at Utö in the coastal region of the Baltic Sea"

_EGUsphere, 2024_

## Referee Comment (RC1)

Title: Interannual and seasonal variability of the air-sea CO2 exchange at Utö in the coastal region of the Baltic Sea
Author(s): Martti Honkanen et al.
MS No.: egusphere-2024-628
MS type: Research article "Interannual and seasonal variability of the air-sea CO2 exchange at Utö in the coastal region of the Baltic Sea"

Hokanen et al. present an interesting article that seeks to quantify CO2 flux between the coastal Baltic Sea waters and the overlying atmosphere across 5 years of observation. Using direct Eddy Covariance measurements across the sea surface from a land-based tower, interspersed with pCO2 gradient measurements, that were incorporated into a wind-driven CO2 flux parameterized model, the investigators were able to estimate annual CO2 flux estimates and report both seasonal flux patterns and interannual variability. Ultimately, they are able to use 5 years of data to estimate CO2 budget for the Archipelago Sea where Utö is located. Importantly, by taking advantage of the long-term environmental monitoring programs and assets at Utö (ICOS Atmospheric station, Marine Research Station, etc.), the authors were able to record from a suite of co-located instruments to 1) refine their pCO2 gradient and flux measurements, 2) suggest plausible forcing factors to explain seasonal and inter-annual variation in pCO2 in water and CO2 flux direction and magnitude. I believe this work should make an important contribution to quantifying CO2 flux in this region of the Baltic Sea and increase overall knowledge, dynamics, and forcing that take place in nearshore ocean ecosystems. The authors address the possible actions of biological, chemical, and physical factors to explain patterns of nutrient input and uptake, suggesting land-sea interaction and lateral transport of nutrients, combined with temperature and solar input as driving seasonal phytoplankton dynamics as a key driver of seasonal CO2 drawdown in the water and wind as especially important in winter months for enhancing gas exchange.

Substantive Concerns
1) Although details of CO2 flux measurements and modeling were detailed in Appendix A (Lines 430-478), I was surprised that some of this information was not included in the main manuscript, e.g., Methods and Results. Perhaps I am missing or misinterpreting (forgive me if that is the case), but it seems that the opportunity to use the direct comparison of EC measurements and the parameterized air-sea CO2 flux results to help validate the flux model is being lost. Although parameterized air-sea CO2 flux data are used for gap filling when accurate EC was disrupted by ships or otherwise not possible, is there a reason not to compare time periods when both EC and pCO2 gradients were conducted to ground-truth model results? This seems especially important since EC, the direct measurement was only used 26% of the time (Line 161). In some sense, it seems that the EC was actually filling gaps for flux model results. Can you comment on this?

2) Figure 9 contains especially important data describing 5 year's of CO2 fluxes observed at Utö, as measured by EC and estimated by parameterized models according to EQ 1 (line

44). However, I think it would be helpful to identify and differentiate EC values from the modeled value in Figure 9.

3) 2.2 Flow-through system (Lines 104-111). Is it possible to report the diameter of the 250m pipe that connects the submerged borehole pump (with inlet at 4.5m depth) to the manifold that feeds the instruments in the station? Although a flow rate of 55 l min-1 is reported, it is not possible to determine the residence time of the water inside the transfer pipe. Additionally, although there was frequent and automatic cleaning of the sensors and instrument hoses (Lines 110-111) there is no information on how frequently 250m pipe is and whether biofouling inside this pipe is likely to affect pCO2 value measured at equilibrator inside the marine station. Lines 131-134: The authors do cite information about potential magnitude of possible pipe effect "The effect of the long inlet tube on the pCO2 measurement at Utö has also been verified to be small (Honkanen et al. 2021), but a brief summary statement about this potential measurement error would be helpful.

4) Figure A1 (page 31). Given the centrality of the gas transfer velocity estimates to CO2 flux, which appear to have been relied on for 74% of observations, I am surprised that this key parameterization result is buried in the Appendix. Given the CO2 flux is the basis for generation of annual CO2 budgets, I wonder if the authors might consider incorporating this key finding and a discussion of it in the main text? If manuscript space/length is an issue, some of the environmental measurement figures (Figs 3-10) could be moved to Supporting Materials, but discussed and referenced in the main body of the text.

Minor suggested edits/comments
Abstract (line 40), please define FCO2 as CO2 flux to avoid confusion with fugacity.

It would be helpful to include the salinity range of the Archipelago Sea in description of study site. General comments about oceanographic and possibly land-sea interactions of the Gulf of Finland and Gulf of Bothnia with Archipelago Sea would be helpful to readers less familiar with region.

Figures
Some figures are positioned in the main text (Figs. 1 and 2) but the other figs appear after the text and Appendix section on pages 23-28, these should be inserted at appropriate locations within the main text of the manuscript.

Figure 2 indicates that pH was one of the measurements made with the flow-through systems, but investigators did not seem to mention whether pH tracked pCO2 and would be expected.

Figure 4 is missing units on the left side vertical axis.

Figure 8 is missing vertical axis label.

Line 151 and elsewhere, (m.a.s.l.), please define acronym (meters above surface level?)

Line 155, "The measured EC fluxes were corrected for the flux loss occurring in the system." Please elaborate.

Lines 207-209: "for the most part of the year" - edit to "for most of the year".
"The year of 2018 was the warmest amongst the studied ones" – edit to ". . . warmest of those studied."

Discussion
Line 323: "as a source" – edit to "as a net source"

Lines 325-339: Authors might consider referring to allocthonous vs. autocthonous carbon. This is simply an observation, but in shallow well mixed estuaries, the benthos can be less disconnected from surface waters, with respect to its ability to mineralize organic carbon than it is in this system. In the current study location, strong thermoclines prevent mixing such that benthic respiration appears to have little or no effect compared with the photosynthetic drawdown of $CO_2$ due to summer phytoplankton blooms.

Lines 358-359: "The seawater $pCO_2$ in the summer 2017 remained extraordinarily high, compared to other years." Think about rephrasing as "In comparison with other years, the seawater $pCO_2$ in the summer 2017 remained unusually high and values nearly exclusively above the multi-year mean."

Line 360: ". . . the sink fluxes were accordingly lower than other years." Consider rephrasing to ". . . the sink fluxes were accordingly weaker than other years."

Lines 363- 364: "The mixing of $CO_2$ rich to the surface may have diluted the drawdown surface, thus decreasing the negative fluxes in summer." Consider changing "decreasing" to "lessening".

Line 366: "Due to this, the sea was able to release more carbon dioxide . . . ." Consider rephrasing to "For this reason, we believe the sea was able to release more carbon dioxide . . . ."

---

## Author Comment (AC1)

egusphere-2024-628 | Honkanen et al. (2024): Interannual and seasonal variability of the air–sea CO$_2$ exchange at Utö in the coastal region of the Baltic Sea

Dear anonymous referee #2,

Thank you for your comments. They highly improve the quality of the publication. Especially clarifications on the flux theory and methods were useful.

Below, the referee comments (shown in red) are answered point by point.

The manuscript presents an extensive data-series of valuable data of air-sea exchange using EC technique as well as bulk calculated fluxes. The data shows the seasonal cycle and net excheange over one region in the Baltic Sea and is a valuable analysis. Some significant clarifications are needed to enrure full understanding of the use of the different methods and the accuracy.

Comments: The gas transfer velcocity is controlled bu severla factors (line 45 page 2), this is of particular importance for a coastal site and should be further discussed (see eg Garbet et al ., 2014).

The gas transfer velocity indeed plays a very important role in governing the air-sea CO$_2$ flux, and the processes controlling it should be elaborated. For this reason, a list of processes will be added in L45:

"Several processes affect the efficiency of the gas transfer, such as micro scale wave breaking, turbulence, bubbles, sea spray, rain, waves and surface films (Garbet et al., 2014, p. 56). Typically the gas transfer velocity is parameterized…"

Also, the appropriate reference will be added to the reference list in L549:

"Garbe, C.S, Rutgersson, A., Boutin, J., de Leeuw, G., Delille, B., Fairall, C.W., Gruber, N., Hare, J., Ho, D.T., Johnson, M.T., Nightingale, P.D., Pettersson, H., Piskozub, J., Sahlée, E., Tsai, W., Ward, B., Woolf, D.K., and Zappa, C.J.: Transfer Across the Air-Sea Interface. In: Liss, P.S., Johnson, M.T. (eds), Ocean-Atmosphere Interactions of Gases and Particles, Springer Earth System Sciences, Springer, Berlin, Heidelberg, https://doi.org/10.1007/978-3-642-25643-1_2, 2014"

Page 7, line 160. The choice of accurate data is unclear (26% of the time), is this due to wind directions only, what about fluxes below detection limit and other errors in the data. Is the non-stationarity the only quality control criteria?

The EC flux data was quality controlled by removing data that was (1) during wrong wind directions, (2) non stationary, (3) affected by the ships or (4) measured during maintenance/malfunction of system. The detection limit was not considered in the quality control. In Honkanen et al. (2018), the wind turbulence statistics are discussed. We also briefly looked into the EC fluxes during low flux months, and the EC fluxes seemed mostly to behave nicely as a function of friction velocity (data not shown).

 L159-162 will be modified to describe the amount of data removed due to the wrong wind directions:

egusphere-2024-628 | Honkanen et al. (2024): Interannual and seasonal variability of the air–sea $CO_2$ exchange at Utö in the coastal region of the Baltic Sea

"The EC method was supported by flux gap-filling for those periods, when **reliable** EC data were not available due to flux footprint originating from the island (outside of the 190–350◦ wind sector), non-stationary flux conditions or CO2 molar fraction disturbance by ships passing the flux footprint area. The EC data were considered appropriate for 26% of the time and the remaining periods were gap-filled as described below. **Approximately 60 % of the time, the wind at Utö blows from appropriate direction suitable for the air-sea exchange measurements (Honkanen et al. 2018). Thus, the reminder of the data removal is caused by the nonstationarity, ship interference or instrument maintenance/malfunctions.** Based on…"

Page 8,line 175: The parameteirsation is stated to be site specific, why is this? What makes the parameterisation site specific. it is enerally thought that the transfer coefficient could be generally described in terms of forcing processes.

Many aspects of the coast, such as fetch and bottom geometry, modify the wave field, and thus likely affect the gas transfer. Fetch is an important factor limiting the wind induced waves. In the case of Utö, the closest shoal is approximately 1.4 km west-southwest, and likely the waves can not grow as freely as in open sea areas. Here, we did not analyse in detail the gas transfer as a function of wind direction but might be interesting in future, especially if connected to other measurements.

L174 will be modified accordingly: "This parametrization is a site specific, due to for instance the wind fetch and the bottom geometry, and slightly differs from the one given for the open ocean…"

Line 180: three sources of data: EC, Bulk and reconstructed, it is unclear how much data of each category and the distribution of situations. Does this in some way bias the analysis?

Combining EC fluxes with parameterized fluxes are common practice. The EC and parameterized flux data likely are distributed uniformly. However, the parameterization using reconstructed/interpolated may have been used mostly during the summer time, when the pipe system was offline. The interpolated or reconstructed data was used for less than 15% of the time. Its accuracy was discussed briefly in L454. Typically, during summer time, the seawater pCO2 is low, the absolute gradient between the sea and the water is large, and the relative inaccuracy effect on the calculated flux should be small.

L462 A text showing the net exchange based on the EC flux will be included:
"The comparison between the annual budgets calculated using the parametrization and the EC method is not straightforward as the EC data set contains frequent gaps. However, the average annual air-sea CO2 exchange using only the EC method was 19.6 gC m−2 y−1, which is slightly lower than the estimate received using the parametrization and the EC data together."

section 3.4: How are the large drops in salinity explained?

L244 A brief explanation on the salinity drops will be added:

"Large drops in salinity indicate fresh water intrusion. As they seem to occur during the late winter or early spring, they are possibly a result of melting snow or ice."

section 3.5: How can the variability in Chl-a be explained?

The full picture of the Chl-a development is out of scope this paper, as it is governed by a complex mixture of meteorological, hydrographical and biological processes. Kraft et al. (2022) has studied the algal dynamics at Utö.

A brief description of the algal dynamics will be added in L253:

"The algal dynamics is controlled by multiple factors, such as solar irradiation, wind and temperature, e.g. calm and warm weather favors cyanobacterial growth (Kraft et al. 2021)."

section 3.7: Also the variability in the wind is relevant, this should be discussed.

Wind is an important factor contributing to the gas transfer and requires more attention. The importance of the wind will be added in L272:

"As the gas transfer velocity has typically been parameterized using a quadratic relation of wind speed (Wanninkhof, 2014), small changes from the 5-year average may prove to be important for enhancing or suppressing the air-sea $CO_2$ flux. For instance, calm weather in late 2019 likely restricted the release of $CO_2$ to the atmosphere."

Page 13, line 340: Here upwelling is discussed as one explanation, it would require some further discussion on the relevance and frequency of upwelling at this site.

In L341 a discussion on the relevance and the frequency of mixing processes will be added:

"The annual overturning of the water column plays an important factor bringing the carbon in the surface, to be released to the atmosphere. Similar effect can be generated by the wind-induced upwelling events, which can occasionally occur in the outer Archipelago (Lehmann et al, 2012)."

In L592 the reference will be added:

" Lehmann, A., Myrberg, K. and Höflich, K.: A statistical approach to coastal upwelling in the Baltic Sea based on the analysis of satellite data for 1990-2009, OCEANOLOGIA, 54, 369-393, https://doi.org /10.5697/oc.54-3.369/, 2012.

line 344: What is here meant by atmospheric deposition (of what)?

L344 will be clarified:

"… atmospheric deposition of carbon which constitutes carbon entering the system by precipitation (wet deposition) and from the atmospheric particles (dry deposition)."

Line 352: Uncertainty estimates on these numbers should be discussed. In addition there are several other estimates to compare with, based on observations and/or modelling.

A discussion on the uncertainty will be added in L355:

"The inaccuracies for the annual air-sea $CO_2$ exchanges were within 7.9-17.9 g C m$^{-2}$ y$^{-1}$, indicating an inaccuracy range of approximately 40 % of the total exchange, and thus the positive net exchanges for all years are certain. The inaccuracy arises mostly from the gas transfer velocity parametrization (Appendix A), and attempts to improve the parametrization are needed."

Appendix A: The EC method, Again, it is unclear if the non-stationalrity is enough as quality criteria (what about detection limit).

The EC flux data was quality controlled by removing data that was (1) during wrong wind directions, (2) non stationary, (3) affected by the ships or (4) measured during maintenance/malfunction of system. The detection limit was not considered in the quality control. In Honkanen et al. (2018), the wind turbulence statistics are discussed. We also briefly looked into the EC fluxes during low flux months, and the EC fluxes seemed mostly to behave nicely as a function of friction velocity.

Two different criteria seems to be used for EC data (for budgets and estimates of transfer velocity). This should be further described and discussed.

A clarification will be added in L458:

"For the analysis of the gas transfer velocity, more strict quality control, compared to the general EC flux data, were used in order to robustly analyze the gas transfer velocity as a function of wind speed."

The choice to not include small pCO2 gradients helps to avoid problems of small values in the denominator, when solving the k from the flux equation. In the case of small values in the denominator, small relative inaccuracies can become problematic.

Description of error estimate is unclear.

L472-477 will be rewritten to be clearer:

"The error estimates for the air–sea CO2 fluxes were calculated assuming that the parametrization of the gas transfer velocity is the dominant source of uncertainty. The error estimate for the annual net exchange of air–sea CO2 was obtained by multiplying the annual balance with the median absolute percentage error (between the EC and parameterized flux), weighted by the proportion of parametrized fluxes. The uncertainty for the average net exchange of CO2 was 11.3 gC m−2 y−1.

The error estimates for each year are given in Table 1. The error estimates for the monthly flux averages (shown in Fig. 9) are calculated as a monthly median absolute difference between the EC flux and parameterized flux, weighted by the proportion of parametrized fluxes."

Garbe C S et al 2014 Ocean-Atmosphere Interactions of Gases and Particles eds P Liss and M

Johnson (Springer-Verlag) 55-112

Suggested upwelling literature:

Lehmann, A., Myrberg, K., 2008. Upwelling in the Baltic Sea - A review. J. Mar. Syst. 74, S3–S12. https://doi.org/10.1016/j.jmarsys.2008.02.010.

Lehmann, A., Myrberg, K., Hoflich, K., 2012. A statistical approach to coastal upwelling in the Baltic Sea based on the analysis of satellite data for 1990–2009. Oceanologia 54, 369–393. https://doi.org/10.5697/oc.54-3.369. Norman, M., Parampil, S.R., Rutgersson, A., Sahl´ee, E., Norman, M.,

Parampil, S.R., Rutgersson, A., Sahl´ee, E., 2013. Influence of coastal upwelling on the air–sea gas exchange of CO 2 in a Baltic Sea Basin. Tellus B Chem. Phys. Meteorol. 65, 1–16. https://doi.org/10.3402/tellusb.v65i0.21831.

Zhang, S., L. Wu, J. Arnqvist, C. Hallgrenand A. Rutgersson. Mapping coastal upwelling in the Baltic Sea from 2002 to 2020 using remote sensing data. Int. J. of Appl. Earth Observation and Geoinformatics, Volume 114, November 2022, 103061, https://doi.org/10.1016/j.jag.2022.103061

Thank you for the suggestion of literature.

Best regards,

Martti Honkanen

---

## Author Comment (AC2)

egusphere-2024-628 | Honkanen et al. (2024): Interannual and seasonal variability of the air–sea CO$_2$ exchange at Utö in the coastal region of the Baltic Sea

Dear anonymous referee #1,

Thank you for your comments. They highly improve the quality of the publication, its scientific content, and the readability of the manuscript.

Below, the referee comments (shown in red) are answered point by point.

Title: Interannual and seasonal variability of the air-sea CO2 exchange at Utö in the coastal region of the Baltic Sea Author(s): Martti Honkanen et al. MS No.: egusphere-2024-628 MS type: Research article "Interannual and seasonal variability of the air-sea CO2 exchange at Utö in the coastal region of the Baltic Sea"

Hokanen et al. present an interesting article that seeks to quantify CO2 flux between the coastal Baltic Sea waters and the overlying atmosphere across 5 years of observation. Using direct Eddy Covariance measurements across the sea surface from a land-based tower, interspersed with pCO2 gradient measurements, that were incorporated into a wind-driven CO2 flux parameterized model, the investigators were able to estimate annual CO2 flux estimates and report both seasonal flux patterns and interannual variability. Ultimately, they are able to use 5 years of data to estimate CO2 budget for the Archipelago Sea where Utö is located. Importantly, by taking advantage of the long-term environmental monitoring programs and assets at Utö (ICOS Atmospheric station, Marine Research Station, etc.), the authors were able to record from a suite of co-located instruments to 1) refine their pCO2 gradient and flux measurements, 2) suggest plausible forcing factors to explain seasonal and inter-annual variation in pCO2 in water and CO2 flux direction and magnitude. I believe this work should make an important contribution to quantifying CO2 flux in this region of the Baltic Sea and increase overall knowledge, dynamics, and forcing that take place in nearshore ocean ecosystems. The authors address the possible actions of biological, chemical, and physical factors to explain patterns of nutrient input and uptake, suggesting land-sea interaction and lateral transport of nutrients, combined with temperature and solar input as driving seasonal phytoplankton dynamics as a key driver of seasonal CO2 drawdown in the water and wind as especially important in winter months for enhancing gas exchange.

Substantive Concerns
1) Although details of CO2 flux measurements and modeling were detailed in Appendix A (Lines 430-478), I was surprised that some of this information was not included in the main manuscript, e.g., Methods and Results.Perhaps I am missing or misinterpreting (forgive me if that is the case), but it seems that the opportunity to use the direct comparison of EC measurements and the parameterized air-sea CO2 flux results to help validate the flux model is being lost. Although parameterized air-sea CO2 flux data are used for gap filling when accurate EC was disrupted by ships or otherwise not possible, is there a reason not to compare time periods when both EC and pCO2 gradients were conducted to ground-truth model results? This seems especially important since EC, the direct measurement was only used 26% of the time (Line 161). In some sense, it seems that the EC was actually filling gaps for flux model results. Can you comment on this?

The discussion on the different flux determination methods is very important for the marine carbon and flux communities. The choice to place the information concerning the differences between the EC and parameterized fluxes in the Appendix was done in order to keep the main focus of the paper in the natural CO$_2$ flux variability.

In sense, the EC flux data was used for gap filling the parameterized fluxes. However, the EC data was used for the verification and fine tuning of the parametrization. We formed the site specific gas transfer velocity parameterization in order to confidently fill the gaps in the flux data, but left the deeper study of the gas transfer processes, with possible other auxiliary instrumentation, for the publications in future.

L462 A text showing the net exchange based solely on the EC flux will be included:
"The comparison between the annual budgets calculated using the parametrization and the EC method is not straightforward as the EC data set contains frequent gaps. However, the average annual air-sea CO2 exchange using only the EC method was 19.6 gC m−2 y−1, which is slightly lower than the estimate received using the parametrization and the EC data together."

To not cause misconceptions, the fluxes for the shorter periods were not discussed here. In further publications, it will be possible to study the fluxes and the relevant processes during shorter periods when both flux methods are available.

2) Figure 9 contains especially important data describing 5 year's of CO2 fluxes observed at Utö, as measured by EC and estimated by parameterized models according to EQ 1 (line 44). However, I think it would be helpful to identify and differentiate EC values from the modeled value in Figure 9. The figure 9b is already crowded with information, and thus separating the different methods (EC and parametrization) for each year would overcomplicate the figure, especially when some months possibly contain very few data points. However, the 5-year monthly EC fluxes fits here, which might provide useful information on the relationship between the EC and the parametrization fluxes. Thus, the figure and its caption will be updated.

3) 2.2 Flow-through system (Lines 104-111). Is it possible to report the diameter of the 250m pipe that connects the submerged borehole pump (with inlet at 4.5m depth) to the manifold that feeds the instruments in the station? Although a flow rate of 55 l min-1 is reported, it is not possible to determine the residence time of the water inside the transfer pipe.
The volume of the structure has been estimated to be 357 l, including all parts of the pumping system. The inner diameter of the long part of the water pipe is approximately 3.3 cm. The residence is approximately 6 min, depending on the flow rate.

The residence time will be added in L110:
"The residence time (on average 6 min) depends on the flow rate and is taken into account in the measurement time."

Additionally, although there was frequent and automatic cleaning of the sensors and instrument hoses (Lines 110-111) there is no information on how frequently 250m pipe is and whether biofouling inside this pipe is likely to affect pCO2 value measured at equilibrator inside the marine station.
Information on the pipe and pump changes will be added in L111:
"The submersible pump and the underwater pressure pipe are renewed annually."

Lines 131-134: The authors do cite information about potential magnitude of possible pipe effect
"The effect of the long inlet tube on the pCO2 measurement at Utö has also been verified to be

small (Honkanen et al. 2021), but a brief summary statement about this potential measurement error would be helpful.

L133 will be modified to introduce more information on the system verification:

"The effect of the long inlet tube on the pCO2 measurement at Utö was verified to be small (4.1 µatm) by using several standalone sensors placed in the inlet location and in a tank connected to the manifold (Honkanen et al., 2021).

4) Figure A1 (page 31). Given the centrality of the gas transfer velocity estimates to CO2 flux, which appear to have been relied on for 74% of observations, I am surprised that this key parameterization result is buried in the Appendix. Given the CO2 flux is the basis for generation of annual CO2 budgets, I wonder if the authors might consider incorporating this key finding and a discussion of it in the main text? If manuscript space/length is an issue, some of the environmental measurement figures (Figs 3-10) could be moved to Supporting Materials, but discussed and referenced in the main body of the text.

The site specific parametrization used in the analysis did not result in significantly different results than that would have gotten with using the common Wanninkhof (2014) parametrizations, stated in the Appendix. The figure A1 serves the purpose of verifying the flux parametrization.

Minor suggested edits/comments

Abstract (line 40), please define FCO2 as CO2 flux to avoid confusion with fugacity.

FCO2 can be easily misinterpreted as fugacity, and thus we define the variable of the air-sea CO2 flux as $F_{as.}$

L40 will be modified:

"The $CO_2$ exchange between the atmosphere and the sea ($F_{as}$)…"

L43 This also affects the equation in L43, which will be modified as:

"$F_{as} = k$…"

It would be helpful to include the salinity range of the Archipelago Sea in description of study site. General comments about oceanographic and possibly land-sea interactions of the Gulf of Finland and Gulf of Bothnia with Archipelago Sea would be helpful to readers less familiar with region.

L91 A salinity range and information on the flow patterns will be added to the text:

"The surface salinity at Utö varies typically between 6-7 PSU (Laakso et al. 2018). The flow patterns in the Archipelago Sea contain large variations in time and space (Erkkilä and Kallio, 2004)."

L533 the reference will be added:

Erkkilä, A. and Kalliola, R.: Patterns and dynamics of coastal waters in multi-temporal satellite images: support to water quality monitoring in the Archipelago Sea, Finland, Estuarine, Coastal and Shelf Science, 60, 165–177, https://doi.org/10.1016/j.ecss.2003.11.024/, 2004.

Figures

Some figures are positioned in the main text (Figs. 1 and 2) but the other figs appear after the text and Appendix section on pages 23-28, these should be inserted at appropriate locations within the main text of the manuscript.

This seems to be a feature of the LaTeX template used. We kindly ask the Copernicus publications to look at this subject to direct the figures at their right places.

Figure 2 indicates that pH was one of the measurements made with the flow-through systems, but investigators did not seem to mention whether pH tracked pCO2 and would be expected.
We are working on the pH data and will use it in upcoming publications. The data has required salinity considerations and other quality controlling. For this paper, we wanted to keep the attention on the net CO2 exchange and thus excluded data not directly needed to the purpose. Preliminarily, the pH and pCO2 at Utö show very strong anticorrelation. Due to this strong anticorrelation, the pH served as a quick glance on the quality control for the pCO2 but this QC method was not done routinely here, and thus not introduced.

Figure 4 is missing units on the left side vertical axis.
Figure 4 has the right units (m) shown.

Figure 8 is missing vertical axis label.
Figure 8 is missing the unit (m/s), and this will be added to the left side of the figure.

Line 151 and elsewhere, (m.a.s.l.), please define acronym (meters above surface level?)
L151 The definition of acronym will be added:
"…, using a 9m tall micrometeorological flux tower, 12 m.a.s.l (meters above sea level), on the western shore of the island."

Line 155, "The measured EC fluxes were corrected for the flux loss occurring in the system." Please elaborate.
L155 The flux loss processes will be elaborated:
"…, which mostly originates from the attenuation of CO2 molar fraction fluctuations in the tubing."

Lines 207-209: "for the most part of the year" - edit to "for most of the year".
"The year of 2018 was the warmest amongst the studied ones" – edit to ". . . warmest of those studied."
L207-209 will be modified accordingly:
"The temperature remained up to approximately 2 °C below the 5-year average for most of the year, with the exception of September–October 2017, when the cooling was slightly lower than for other years. The year of 2018 was the warmest of those studied, in terms of the summer peak temperatures."

Discussion
Line 323: "as a source" – edit to "as a net source"
L323 will be clarified:
"The marine ecosystem at Utö acted as a net source of atmospheric carbon dioxide,…"

Lines 325-339: Authors might consider referring to allocthonous vs. autocthonous carbon. This is simply an observation, but in shallow well mixed estuaries, the benthos can be less disconnected from surface waters, with respect to its ability to mineralize organic carbon than it is in this system. In the current study location, strong thermoclines prevent mixing such that benthic respiration appears to have little or no effect compared with the photosynthetic drawdown of CO2 due to summer phytoplankton blooms.
The benthic coupling is an important path way of the carbon. Unfortunately, it is not easily determined at the moment.

L331 will be modified:

"Organic carbon originates primarily from photosynthesis, either locally (autochtonous) or imported from elsewhere (allochtonous)."

In L341 a discussion on the mixing processes will be modified:

"The annual overturning of the water column plays an important factor bringing the carbon in the surface, to be released to the atmosphere. Similar effect can be generated by the wind-induced upwelling events, which can occasionally occur in the outer Archipelago (Lehmann et al, 2012)."

In L592 the reference will be added:

" Lehmann, A., Myrberg, K. and Höflich, K.: A statistical approach to coastal upwelling in the Baltic Sea based on the analysis of satellite data for 1990-2009, OCEANOLOGIA, 54, 369-393, https://doi.org /10.5697/oc.54-3.369/, 2012.

Lines 358-359: "The seawater pCO2 in the summer 2017 remained extraordinarily high, compared to other years." Think about rephrasing as "In comparison with other years, the seawater pCO2 in the summer 2017 remained unusually high and values nearly exclusively above the multi-year mean."
L358-359 were modified accordingly:
"In comparison with other years, the seawater pCO2 in the summer 2017 remained unusually high and values nearly exclusively above the multi-year mean."

Line 360: ". . . the sink fluxes were accordingly lower than other years." Consider rephrasing to ". . . the sink fluxes were accordingly weaker than other years."
L360: ". . . the sink fluxes were accordingly lower than other years." will be reworded to ". . . the sink fluxes were accordingly weaker than other years."

Lines 363- 364: "The mixing of CO2 rich to the surface may have diluted the drawdown surface, thus decreasing the negative fluxes in summer." Consider changing "decreasing" to "lessening".
L363 decreasing will be reworded to lessening.

Line 366: "Due to this, the sea was able to release more carbon dioxide . . . ." Consider rephrasing to "For this reason, we believe the sea was able to release more carbon dioxide . . . ."

L366: "Due to this, the sea was able to release more carbon dioxide . . . ." will be replaced to "For this reason, we believe the sea was able to release more carbon dioxide . . . ."

Best regards,

Martti Honkanen

---

## Author Response (AR1)

EGUSPHERE-2024-628

Dear editor,

I included the modifications suggested by the reviewers. The changes are listed in the referee comments and shown in the track-changes file.

However, one reviewer suggested that the figures should be shown within the text. Now, the result figures come after all the text for some reason. I think this is an issue with the Copernicus Overleaf template, and I possibly require editorial help with this, after I have submitted the LaTeX files.

Best regards,

Martti Honkanen

---

## Editor Decision (ED1)

[revised manuscript text omitted]

**(b) air-sea flux of carbon dioxide**

[Figure]

**Figure 9.** (a) Daily means of the seawater $pCO_2$ at the depth of 4.5 m at Utö in 2017–2021. Interpolated observations are shown with dotted line. (b) Monthly average $CO_2$ air–sea fluxes. The error estimates are explained in the Appendix.

[Figure]

**Figure 10.** Cumulative sum of the monthly means, starting from January 1st, of the air–sea $CO_2$ fluxes at Utö for the years 2017–2021.

[Figure]

**Figure A1.** Gas transfer velocity, normalized to the Schmidt number of 660, as a function of wind speed.

---

## Author Response (AR2)

EGUSPHERE-2024-628

Dear editor,

Thank you for the comments and editorial work on this publication. I have included all the modifications suggested. The changes are listed below.

L36 "mobilization" was replaced by "export".

L38 "atmospheres" was modified to "atmosphere".

L38 "carbon sequestration" was modified to "oceanic carbon sequestration".

Fig. 2 caption was modified to include the explanations of all the abbreviations:

"TD denotes a standalone temperature-depth sensor, attached to the inlet. CDOM stands for colored dissolved organic matter and $O_2$ stands for dissolved oxygen."

L308 The extra dash was removed from the "spring—summer"

L363 "remain" was modified to "remains"

L370, L371, L374, L377 The unit was corrected from " $gC\ m^{-2}\ s^{-1}$ " to " $gC\ m^{-2}\ y^{-1}$ "

L371 The subscript in "CO2" was corrected to "$CO_2$"

L526 An acknowledgment for the referees and editor was added:

"We thank two anonymous referees and the associate editor Frédéric Gazeau for their comments, which improved the publication."

Fig. 4 dots denoting the thermocline depths were recolored from black to white to increase the visibility. Also, the abbreviation of CTD was explained in the caption:

"Temperature vertical profiles (colours) and the thermocline depths (white dots) at Utö for the years 2017-2021, derived from the conductivity-temperature-depth casts."

Fig. 6 "Chl-a" was modified to "Chlorophyll-a" in the caption.

Fig. 8 The "winds peed" in the caption was corrected to "wind speed".

Fig. 9 The "$p\text{CO}_2$" was replaced by "$\text{CO}_2$ partial pressure". Also, the different flux values were explained:

"(a) Daily means of the $\text{CO}_2$ partial pressure in the seawater at the depth of 4.5 m at Utö in 2017-2021. (b) Monthly average $\text{CO}_2$ air-sea fluxes: the 5-year average fluxes constructed using both eddy covariance and parametrization (black), the 5-year average eddy covariance fluxes (gray) and the fluxes for different years using both eddy covariance and parametrization (other colours). The error estimates are explained in the Appendix."

Best regards,

Martti Honkanen